# Beyond Hamming: Query-Aware Decoding of Binary Cosine Sketches

DaeHun Nyang [1]

## Abstract

Cosine similarity estimation is a core primitive in coarse-to-fine retrieval pipelines, where early-stage candidate selection relies on approximate similarity estimates whose errors are amplified downstream. Widely used sign-based sketches arising from extreme quantization of random projections exhibit a structural variance peak near $\theta \approx 90°$, the near-background region where candidate selection is most difficult. We propose QA-COS, a query-aware decoder-side estimator that departs from the Hamming-agreement paradigm, treating sign bits as probabilistic observations rather than deterministic votes. Across simulations and BEIR benchmarks, QA-COS reduces estimation error by up to $\sim$15–20% in the near-orthogonal region and translates these gains into improved candidate selection in two-stage ANN pipelines, improving Hit@K by up to $\sim$30 percentage points at fixed budgets and reducing candidates by up to $\sim$45–50% at fixed recall. In a native `hnswlib` scorer-replacement experiment, the same decoder improves the quality of the final HNSW frontier, and in a focused storage-aware setting a practical gated variant reduces end-to-end wall-clock latency at matched recall.

## 1. Introduction

Cosine similarity between high-dimensional vectors underpins a wide range of retrieval systems and serves as the de facto scoring function for dense text retrieval pipelines (Manku et al., 2007; Nogueira & Cho, 2019; Reimers & Gurevych, 2019; Karpukhin et al., 2020; Khattab & Zaharia, 2020). Beyond retrieval, cosine similarity also plays a central role in learning objectives such as contrastive representation learning (Chen et al., 2020), as well as in large-scale recommendation systems (Covington et al., 2016).

[1]Division of AI and Software, Ewha Womans University, Seoul, Korea. Correspondence to: DaeHun Nyang <nyang@ewha.ac.kr>.

*Proceedings of the 43rd International Conference on Machine Learning*, Seoul, South Korea. PMLR 306, 2026. Copyright 2026 by the author(s).

**Scope: decoder-side cosine estimation for binary sketches.** At scale, retrieval systems often use a coarse-to-fine strategy: a fast candidate-generation stage operates under strict memory and latency constraints, and a later stage rescores a smaller set with full-precision vectors (Johnson et al., 2017; Guo et al., 2020; Nogueira & Cho, 2019). The coarse stage may be implemented with graph indexes, inverted files, learned or codebook-based quantizers, binary sketches, or combinations of these components (Milvus Team, 2025; Facebook AI Research, 2025; Jégou et al., 2011; Johnson et al., 2017; Malkov & Yashunin, 2016; Lin et al., 2021; Ma et al., 2023). This paper focuses on one specific primitive within this broader design space: estimating cosine similarity from a fixed binary random-projection sketch of a database vector and a real-valued query. We do not aim to replace full ANN indexing pipelines or trained vector-compression methods. Instead, we ask whether an existing sign sketch can be decoded more accurately without changing the stored database codes, retraining a quantizer, or rebuilding the index.

**Why this primitive matters.** Binary cosine sketches remain attractive because they are data-independent, compact, and self-contained: each database item carries its own code, and the standard decoder reduces to a Hamming-agreement count (Charikar, 2002). These properties make sign sketches useful as a coarse filtering primitive, as a lightweight reranking signal inside larger ANN systems, or as a storage-constrained fallback when auxiliary codebooks and lookup tables are undesirable. In these settings, errors in the coarse similarity estimate are amplified into system-level effects: a conservative threshold inflates candidate sets and I/O, while an aggressive threshold drops relevant items before exact rescoring can recover them.

**Sign-only sketches: attractive, yet least reliable near the background.** Sign-only sketches based on random projections provide a structurally simple and widely used approach to similarity estimation. At a more abstract level, the Johnson-Lindenstrauss lemma guarantees that random linear projections can reduce dimensionality while preserving pairwise distances and angles (Johnson & Lindenstrauss, 1984). Goemans and Williamson showed that under sign-based random hyperplane rounding, the probability of separating two vectors is proportional to angular distance (Goemans & Williamson, 1995). SimHash can be interpreted as a canoni-

cal instantiation of this lineage, translating angular similarity into Hamming agreement, and thereby fixing decoding to deterministic bit counting (Charikar, 2002). However, this Hamming-agreement induces angle-dependent uncertainty: bit agreement becomes maximally noisy near $\theta \approx 90°$ (Appendix G). As a result, the decoder exhibits peak variance in this region, precisely where coarse estimates are most critical for candidate generation. This peak does not arise from random projection itself, but from how information discarded through binarization is handled at inference time. Most prior work mitigates this uncertainty by refining the *encoding* process or altering candidate generation strategies.

**This work: query-aware decoding for binary cosine sketches.** While database vectors are stored only through binary sketches, the query remains available as a real-valued embedding at inference time. This decoder-side asymmetry raises a simple but fundamental question: *why should the query be treated as binary?* Building on this asymmetry, we propose QA-COS, a **Q**uery-**A**ware **Cos**ine estimator that *departs from the Hamming-agreement decoding paradigm*, treating sign bits as probabilistic observations rather than deterministic votes during decoding.

**Contributions.**

- **Problem framing (near-background is the bottleneck).** We define the scope of QA-COS as decoder-side cosine estimation from fixed binary random-projection sketches, and identify the near-background region ($\theta \approx 90°$) as the bottleneck where sign-only Hamming decoding has maximal variance and retrieval thresholds are most fragile.

- **Decoder-side, database-preserving enhancement.** We propose QA-COS that keeps the stored binary codes and projection hyperplanes unchanged, while improving *decoding* by exploiting query-side projection magnitudes, yielding a decoder-side estimator that is orthogonal to encoder-side improvements and requires no training or re-indexing.

- **Theoretical formulation and guarantees for query-aware decoding.** We derive a query-conditioned probit likelihood for 1-bit random projections, providing a probabilistic alternative to Hamming-agreement decoding. We show that the resulting 1D MLE is well-posed and admits a unique maximizer under mild conditions.

- **System-level gains under tight two-stage retrieval budgets.** We demonstrate that the accuracy gains of QA-COS translate into system-level improvements in a two-stage retrieval pipeline on BEIR (ArguAna, FiQA-2018, NFCorpus, SciFact), consistently increasing Hit@K while reducing the candidate budget required for hard (near-background) queries. We further validate the same decoder as a search-time scorer replace-

ment inside native `hnswlib`, and report a storage-aware wall-clock study showing that a gated practical variant can reduce end-to-end latency at matched recall.

## 2. Decision Boundaries in the Background

**Motivation.** Candidate generation is most sensitive when the retained frontier lies close to the query-specific background score distribution: a small coarse-estimation error can either inflate the candidate set or drop items before exact rescoring. We separate two cases. If useful items are already below the background frontier in the original float-embedding space, cosine itself is the bottleneck and no binary-sketch decoder can recover them. Our target is the complementary *small-margin but recoverable* region, where float cosine places useful candidates near the retained frontier but sign-only decoding may perturb their order. Figure 1 gives relevance-based evidence that useful items can lie near this frontier and shows the complementary sign-only error peak; the analysis below removes relevance labels and asks where top-$k$ frontiers sit geometrically.

**Pure-geometry frontier evidence.** Let $\gamma_{q,k}$ be the exact top-$k$ cosine cutoff over the corpus and $\tau_{q,\alpha}$ the query-specific $(1 - \alpha)$ background-tail threshold, both computed from float scores without qrels. Figure 2 shows that, across ALL-MINILM-L6-V2 and ALL-MPNET-BASE-V2, the top-$k$ frontier is often close to the $\alpha = 10^{-3}$ background tail. This is regime evidence, not a semantic sufficiency claim: it identifies where a coarse estimator must preserve small differences near a query-specific frontier.

**Why extreme 1-bit quantization is brittle near orthogonality.** Cosine estimators based on sign-only (1-bit) random projections are intrinsically least reliable in the near-background region ($\rho \approx 0$). Formally, for random-hyperplane sign observations, the collision identity $p = 1 - \theta/\pi$ implies $p \approx \frac{1}{2}$ at $\theta \approx \pi/2$, where the Bernoulli variance $p(1 - p)$ is maximized; a delta-method calculation yields the corresponding variance peak in the cosine estimate (Appendix G). Empirical bin-wise curves in Figure 1(d) show the same near-zero error peak.

**A mismatch between operating region and sign-only estimation.** The two observations above meet at the same point. High-dimensional normalized embeddings place most random background items near $\rho \approx 0$, and the BEIR measurements show that many queries have relevant items close to this background tail under tight candidate budgets. Meanwhile, the Hamming decoder for sign sketches has its largest variance in the same near-orthogonal region. This creates a query-dependent ranking problem rather than a uniform estimation problem: near a top-$k$ frontier, many

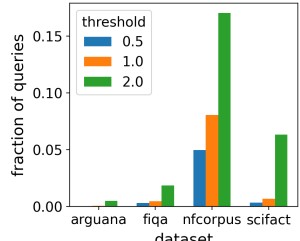 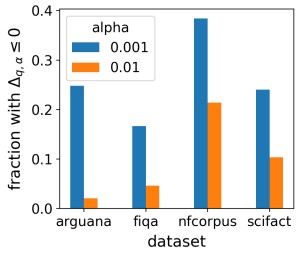 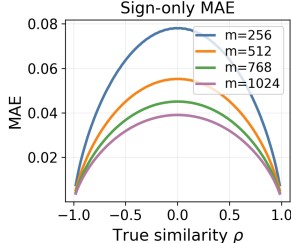

*Figure 1.* **Motivation evidence for near-background decoding.** (**a**) Empirical CDFs of the normalized relevance margin $m_q = (s_q^+ - \mu_q)/\sigma_q$. (**b**) Fraction of queries below fixed margin thresholds. (**c**) Fraction with $\Delta_{q,\alpha} = s_q^+ - \tau_{q,\alpha} \leq 0$, showing that useful items can lie near the query-specific background frontier. (**d**) Mean absolute error of the Hamming/SimHash cosine estimator, binned by true similarity; the error peaks near $\rho \approx 0$. Panels (a)–(c) are qrels-dependent diagnostics, while panel (d) is a sign-only estimator diagnostic.

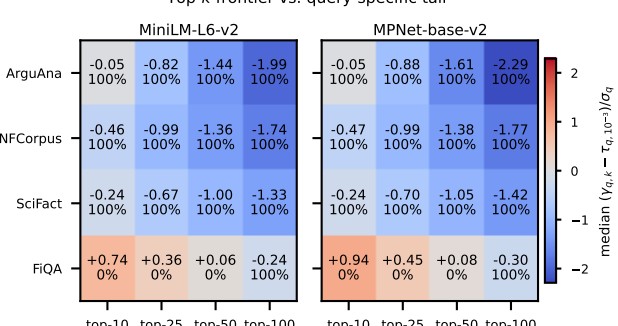

*Figure 2.* **Pure-geometry regime evidence from original float embeddings.** For each query, the exact top-$k$ cutoff in the original dense embedding space is compared with the query-specific background tail $\tau_{q,10^{-3}}$ at $\alpha = 10^{-3}$, independently of relevance labels. We report $k \in \{10, 25, 50, 100\}$ for ALL-MINILM-L6-V2 and ALL-MPNET-BASE-V2. Each cell reports the median normalized gap $(\text{cutoff}_{q,k} - \tau_{q,10^{-3}})/\sigma_q$ and the percentage of queries whose top-$k$ cutoff is at or below the tail. This is regime evidence only: it shows where realistic top-$k$ frontiers sit relative to the background distribution, independent of qrels or ANN outcomes. Direct utility evidence is reported in the retrieval and native HNSW experiments in section 5.

background and useful items can have cosine scores separated by only a small fraction of the local background scale. In this region, unbiased but high-variance sign-only estimates are enough to swap the order of borderline candidates, and these local inversions are amplified by hard budget or threshold decisions before exact rescoring is applied. Thus, the practical failure mode is not merely that binary sketches are coarse; it is that their standard decoder is least stable precisely where candidate selection is most margin-sensitive. The same mismatch can appear as inflated candidate sets when thresholds are relaxed to avoid false negatives, or as missed positives when budgets are kept fixed.

**An opportunity at the decoding stage.** Because the database code is fixed but the query is available in full precision, this mismatch can be addressed at decoding time. QA-COS keeps the stored sign sketch unchanged and mod-

els each database sign bit as a query-conditioned probabilistic observation. The resulting one-dimensional MLE uses query-side projection magnitudes to downweight ambiguous bits and extract more information from confident ones, reducing uncertainty in the near-background region without changing the encoder, training data, or index.

## 3. From Geometry to a Tractable Likelihood

**Problem setting and geometric view.** Angular similarity based on random projections is scale-invariant, so only the direction of a vector matters. We therefore normalize all vectors and work on the unit sphere $\mathbb{S}^{d-1} = \{x \in \mathbb{R}^d : \|x\|_2 = 1\}$, with similarity measured by $\rho = \cos\theta = A^\top B$. We consider a database vector $A \in \mathbb{S}^{d-1}$ that is *latent* at query time: instead of storing $A$ itself, we retain only a sign-only sketch $c_A \in \{\pm 1\}^m$, obtained by extremely quantizing random linear projections with hyperplane normals $H = \{h_i\}_{i=1}^m$, as in random-hyperplane hashing (SimHash). In the probabilistic model we take $h_i \overset{\text{iid}}{\sim} \mathcal{N}(0, I_d)$; for the geometric view only the induced directions matter, since signs are invariant to positive rescaling of $h_i$. Given a real-valued query vector $B \in \mathbb{S}^{d-1}$, our goal is to estimate the cosine similarity $\rho = \cos\theta$ using $c_A$ with query-side information. For contrast, Appendix A illustrates the symmetric uncertainty when both database and query vectors are observed only through binary sketches.

**Notation.** For each hyperplane $i \in [m]$, define the latent database projection $a_i = h_i^\top A$, the observed query projection $x_i = h_i^\top B$, the stored database bit $s_i = \text{sign}(a_i) = c_A[i]$, the query bit $y_i = \text{sign}(x_i) = c_B[i]$, and the query magnitude $b_i = |x_i|$. The empirical Hamming agreement rate is

$$\hat{p} = \frac{1}{m}\sum_{i=1}^m \mathbb{I}\{s_i = y_i\}, \qquad \hat{\rho}_0 = \cos\big(\pi(1 - \hat{p})\big).$$

The query-aware decoder observes $(s_i, x_i)$, equivalently $(s_i, y_i, b_i)$, while the standard Hamming decoder observes only the agreement events $\mathbb{I}\{s_i = y_i\}$.

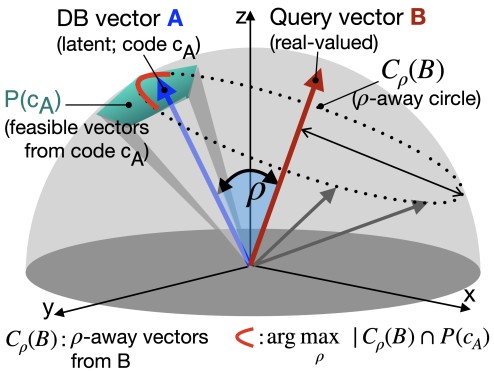

*Figure 3.* **Geometric interpretation of QA-COS on the unit sphere.** The database item is a latent real-valued vector $A$, observed only through its SimHash code $c_A$, which induces a feasible region $P(c_A) \subset \mathbb{S}^{d-1}$ (**the green patch**) on the sphere. Given a real-valued query vector $B$, the set of vectors with cosine similarity $\rho$ to $B$ forms $\mathcal{C}_\rho(B)$, a small sphere centered around $B$ (**the dotted circle**). For a fixed $\rho$, **the red arc** shows the intersection between the $\rho$-circle around $B$ and the patch $P(c_A)$; QA-COS selects the $\rho$ maximizing this overlap. See Appendix A for a complementary view under fully binary representations.

**Candidate vectors of $A$ as a spherical patch.** Geometrically, each bit $c_A[i] = \operatorname{sign}(h_i^\top A)$ restricts $A$ to a hemisphere of $\mathbb{S}^{d-1}$, and the intersection of these constraints defines a feasible spherical patch

$$P(c_A) \triangleq \{x \in \mathbb{S}^{d-1} : \operatorname{sign}(h_i^\top x) = c_A[i] \ \forall i \in [m]\}. \tag{1}$$

**Candidate angles as circles around $B$.** For a candidate cosine similarity $\rho$, the set of all unit vectors with cosine $\rho$ to the query $B$ forms a *spherical circle* centered at $B$:

$$\mathcal{C}_\rho(B) \triangleq \{x \in \mathbb{S}^{d-1} : x^\top B = \rho\}. \tag{2}$$

Figure 3 illustrates this geometry, with the green patch on the sphere denoting the feasible set $P(c_A)$, the brown arrow indicating the query vector $B$, and the dotted circle centered at $B$ denoting $\mathcal{C}_\rho(B)$. Sweeping $\rho$ expands/shrinks this circle centered at $B$.

**From geometric overlap to a tractable likelihood.** Geometrically, cosine estimation can be viewed as choosing $\rho$ whose $\rho$-circle has the longest arc (the red arc) with $P(c_A)$:

$$\widehat{\rho} \in \arg\max_{\rho \in [-1, +1]} |\mathcal{C}_\rho(B) \cap P(c_A)|, \tag{3}$$

where $|\cdot|$ denotes the intrinsic measure on the $(d-2)$-dimensional small-sphere manifold. Directly computing this overlap is intractable, however, since $P(c_A)$ is defined by the intersection of $m$ hemispheres. Instead, we optimize a bit-level likelihood that is consistent with the same geometry: as $\rho$ varies, candidate points on $\mathcal{C}_\rho(B)$ cross hyperplane boundaries, changing the probability of observing each sign bit $c_A[i]$. This perspective reduces angle estimation to maximizing a one-dimensional likelihood in $\rho$.

---

**Algorithm 1** QA-COS: Query-aware cosine estimation

**Require:** Query $B \in \mathbb{S}^{d-1}$; database bits $c_A \in \{\pm 1\}^m$; hyperplanes $H = \{h_i\}_{i=1}^m$; Newton steps $T$; damping $\eta \in (0, 1]$; stability constants $\varepsilon_1, \varepsilon_2$.

**Ensure:** Estimated cosine $\widehat{\rho} \approx A^\top B$.
  *// Project query and form its SimHash bits*
  1: **for** $i \leftarrow 1$ **to** $m$ **do**
  2:     $x_i \leftarrow h_i^\top B$
  3:     $c_B[i] \leftarrow \operatorname{sign}(x_i)$ *// ties occur with probability zero.*
  4: **end for**
  *// Baseline: Hamming agreement $\to$ cosine*
  5: $\widehat{p} \leftarrow \frac{1}{m} \sum_{i=1}^m \mathbf{1}\{c_A[i] = c_B[i]\}$
  6: $\widehat{\rho}_0 \leftarrow \cos(\pi(1 - \widehat{p}))$
  *// Exclude extreme-confidence cases for numerical stability*
  7: **if** $\min(\widehat{p}, 1 - \widehat{p}) \leq \frac{\varepsilon_1}{m}$ **then return** $\widehat{\rho}_0$
  8: **end if**
  *// Probit MLE inside the gate with unconstrained $t \in \mathbb{R}$*
  9: $s_i \leftarrow c_A[i]$ for $i = 1, \dots, m$
  10: $t \leftarrow \operatorname{arctanh}(\operatorname{clip}(\widehat{\rho}_0, -1 + \varepsilon_2, 1 - \varepsilon_2))$
  11: **for** $k \leftarrow 1$ **to** $T$ **do**
  12:     $t \leftarrow t - \eta \cdot \frac{\ell'(t)}{\ell''(t)} \{\ell(t) = \sum_{i=1}^m \log \Phi(s_i x_i \sinh t)\}$
  13: **end for**
  14: $\widehat{\rho} \leftarrow \tanh(t)$
  15: **return** $\widehat{\rho}$

---

## 4. Query-Aware Cosine Estimation

We propose a 1D MLE that improves cosine estimation by optimizing a likelihood consistent with the geometric overlap between $P(c_A)$ and the $\rho$-circle.

**Setup.** We use the notation introduced in Section 3. In particular, $A, B \in \mathbb{S}^{d-1}$, $\rho := A^\top B \in (-1, 1)$, $h_i \overset{\text{iid}}{\sim} \mathcal{N}(0, I_d)$, and $(a_i, x_i, s_i, y_i, b_i)$ denote the database projection, query projection, stored database bit, query bit, and query magnitude, respectively. The database stores only $s_i = c_A[i]$, while the query can compute $x_i$ and $b_i = |x_i|$ for all $i$. Let $\Phi(\cdot)$ and $\phi(\cdot)$ denote the CDF and PDF of the standard normal distribution.

**Query-aware cosine estimation.** Algorithm 1 estimates $\rho = A^\top B$ by selectively refining it near the background. In this ambiguous region, QA-COS maximizes a one-dimensional probit log-likelihood using query-side projection magnitudes, yielding an $O(m)$-per-iteration decoder-side enhancement. The optimization is carried out in an unconstrained parameterization $\rho = \tanh t$, initialized from the baseline estimate to provide a stable warm start and solved with a small number of damped Newton steps. For numerical stability, QA-COS is applied only when $\min(\widehat{p}, 1 - \widehat{p}) > \varepsilon_1/m$, since for $\min(\widehat{p}, 1 - \widehat{p}) = O(1/m)$ the Hamming estimator is already near-deterministic. We

additionally clip the initialization $\widehat{\rho}_0$ to $[-1 + \varepsilon_2, \, 1 - \varepsilon_2]$ to avoid boundary issues in the $\tanh$ reparameterization.

**Algorithmic reproducibility.** The decoder has no trained parameters or data-dependent calibration. Main experiments use $T = 2$, $\eta = 1$, $\varepsilon_1 = 3$, and $\varepsilon_2 = 0.999$ with a fixed hyperplane seed per $(d, m)$ configuration. Database pre-processing stores only $c_A = \text{sign}(HA)$; at query time, $x = HB$ and $c_B = \text{sign}(x)$ are reused by both SimHash and QA-Cos. The 1D Newton objective is initialized from the Hamming cosine estimate and evaluated with the closed-form derivatives in Lemma 4.4, using stable Gaussian log-CDF/log-PDF evaluations for the inverse Mills ratio.

### 4.1. Model: Why a probit likelihood follows

**Probabilistic view of sign observations.** In contrast to Hamming-based decoding, which treats each bit as a uniform vote, our formulation views sign bits as noisy measurements whose reliability depends on the query. To exploit projection magnitudes $\{b_i\}$, we establish a principled relationship between the signs $\{s_i\}$ and the cosine similarity $\rho$. Under random-hyperplane hashing, each bit results from thresholding a real-valued projection, and its reliability depends on the confidence of that projection captured by $b_i$. Lemma E.1 provides a Gaussian conditional model for projections given the query, and applying the sign observation yields the query-aware probit likelihood in Lemma 4.1.

**Probit link and query-aware weighting.** We adopt a *probit* observation model (McCullagh & Nelder, 1989; Albert & Chib, 1993), which provides a convenient probabilistic interpretation of sign observations. From the bivariate normality of projections (See Lemma E.1 in Appendix), $a_i \mid x_i \sim \mathcal{N}(\rho x_i, \, 1 - \rho^2)$. Equivalently, $a_i = \rho x_i + \sqrt{1 - \rho^2}\, \varepsilon_i$ with $\varepsilon_i \sim \mathcal{N}(0, 1)$, and $s_i = \text{sign}(a_i)$, yielding a probit likelihood. To obtain a smooth likelihood in $\rho$ while remaining faithful to the fixed sign observations, we interpret each bit through an equivalent probit decoder, which naturally exposes query-dependent informativeness via $x_i = h_i^\top B$. Here and below, "informativeness" refers to Fisher information for the scalar parameter $\rho$: a bit is more informative when its conditional Fisher contribution to estimating $\rho$ is larger.

**Lemma 4.1** (Query-aware probit likelihood). *Let* $x_i := h_i^\top B \in \mathbb{R}$ *and* $s_i := \text{sign}(h_i^\top A) \in \{\pm 1\}$. *Then*

$$\Pr(s_i \mid x_i, \rho) = \Phi\left( \frac{s_i \, \rho \, x_i}{\sqrt{1 - \rho^2}} \right). \tag{4}$$

*For fixed $\rho$, the probit argument has magnitude* $|\rho|\, b_i / \sqrt{1 - \rho^2}$, *where* $b_i = |x_i|$.

*Proof in Appendix.* See Appendix F.2.

Lemma 4.1 shows that each bit is a probit observation whose

conditional signal-to-noise argument scales with $b_i$: large $b_i = |x_i|$ makes the observed sign more sensitive to $\rho$. This stands in contrast to Hamming-based estimation, which weights all bits equally. See Appendix F.1 for the geometric intuition. A sharper quantitative characterization follows from the Fisher information analysis: Theorem F.2 in Appendix F.4 shows that the conditional Fisher information decomposes across bits, with each bit contributing on the order of $b_i^2$ (see Appendix F.4). Interestingly, this query-aligned weighting principle is closely related to the intuition underlying SCaNN and SOAR; we elaborate on this connection in Appendix D.

**From likelihood to an estimator.** Given Lemma 4.1, the natural estimator for $\rho$ is maximum likelihood. Reparameterizing $\rho = \tanh t$ removes the constraint and yields a smooth, unconstrained one-dimensional objective (Appendix F.3).

### 4.2. Statistical and computational guarantees

**Statistical guarantee: magnitudes are information-non-decreasing.** QA-Cos uses the query-side real-valued projections, rather than relying on sign-only information. This is principled rather than heuristic: observing $x_i$ refines observing only its sign $y_i := \text{sign}(x_i)$, so it cannot decrease Fisher information about the unknown similarity parameter. For an observation $O$ with density or mass function $f_\rho(o)$, we write

$$\mathcal{I}(\rho; O) \triangleq \mathbb{E}_{O \sim f_\rho}\left[ \left( \frac{\partial}{\partial \rho} \log f_\rho(O) \right)^2 \right],$$

assuming the support of $O$ does not depend on $\rho$, the score exists, and differentiation may be interchanged with integration or summation. For multiple independent observations, Fisher information is additive.

**Theorem 4.2** (Information monotonicity under observation refinement). *Consider the probit observation model* $\Pr(s_i \mid x_i, \rho)$ *from Lemma 4.1. Since* $(s_i, y_i = \text{sign}(h_i^\top B))$ *is a deterministic function of* $(s_i, x_i)$, *the data-processing inequality for Fisher information (proved in Appendix F.5) gives*

$$\begin{aligned} \mathcal{I}(\rho; \, s_i, x_i) &\geq \mathcal{I}(\rho; \, s_i, y_i), \\ \mathcal{I}(\rho; \, s_{1:m}, x_{1:m}) &\geq \mathcal{I}(\rho; \, s_{1:m}, y_{1:m}), \end{aligned} \tag{5}$$

*under the regularity conditions stated above. Equivalently, because* $x_i$ *and* $(y_i, b_i)$ *contain the same information, augmenting sign-only decoding with query-side magnitudes cannot reduce information:* $\mathcal{I}(\rho; \, s_i, y_i, b_i) \geq \mathcal{I}(\rho; \, s_i, y_i)$, *and likewise for* $1{:}m$.

*Proof in Appendix.* See Appendix F.5.

Theorem 4.2 formalizes the intuition that incorporating query-side magnitudes is *never less informative* than sign-only observations.

**Computational guarantee: well-posed 1D objective and efficient solvers.** Beyond statistical gains, QA-COS provides a concrete computational guarantee: estimation reduces to maximizing a *1D* probit log-likelihood. In particular, the objective is well-posed under a mild non-separability condition, ruling out the complete-separation pathology of generalized linear models and ensuring a unique maximizer.

**Theorem 4.3** (Existence and uniqueness of the MLE (non-separable case))**.** *Let $z_i := s_i x_i$ and assume there exist indices $i, j$ such that $z_i > 0$ and $z_j < 0$. Then $\ell(t) = \sum_{i=1}^m \log \Phi(z_i \sinh t)$ has a unique maximizer $\hat{t} \in \mathbb{R}$, hence a unique MLE $\hat{\rho} = \tanh \hat{t}$.*

*Proof in Appendix.* See Appendix F.6.

Consequently, the MLE can be computed robustly using standard one-dimensional solvers (e.g., damped Newton). To instantiate it, we next provide closed-form of derivatives.

**Lemma 4.4** (Derivatives of the query-aware probit log–likelihood (in $t$))**.** *Let $x_i \in \mathbb{R}$ and $s_i \in \{\pm 1\}$. Define $z_i(t) = s_i x_i \sinh t$, and let $\lambda(z) = \phi(z)/\Phi(z)$. Then*

$$\ell'(t) = \sum_{i=1}^m \lambda(z_i(t)) \, z_i'(t),$$

$$\ell''(t) = \sum_{i=1}^m \Big[ \lambda'(z_i(t))\big(z_i'(t)\big)^2 + \lambda(z_i(t)) \, z_i''(t) \Big], \tag{6}$$

*where $z_i'(t) = s_i x_i \cosh t$, $z_i''(t) = s_i x_i \sinh t$, and $\lambda'(z) = -\lambda(z)\big(z + \lambda(z)\big)$.*

*Proof in Appendix.* See Appendix F.7.

We apply a damped Newton update with a fixed damping factor $\eta$, using second-derivative clipping to maintain numerical stability and a reliable ascent direction.

**Corollary 4.5** (Damped Newton update for Probit MLE (in $t$))**.** *Under the setting of Lemma 4.4, a damped Newton–Raphson step for maximizing $\ell(t)$, computed using the Gaussian Mills ratio, is*

$$t \;\leftarrow\; t - \eta \, \frac{\ell'(t)}{\ell''(t)}, \qquad \eta \in (0, 1], \tag{7}$$

*where $\ell'(t)$ and $\ell''(t)$ are given in Lemma 4.4. Each iteration computes $\ell'(t)$ and $\ell''(t)$ by a single pass over $i = 1, \ldots, m$, yielding $O(m)$ time per iteration.*

A small fixed iteration budget is sufficient in practice: the variance diagnostic in Section 5.1 shows that $T = 1$ is already close to $T = 2$, and that additional steps beyond $T = 2$ do not materially improve the angle-estimation curve. We therefore use $T = 2$ unless a lower-latency setting explicitly uses $T = 1$.

**Speed.** QA-COS adds a small decoder-side cost (tens of $\mu$s per refined candidate under the reproducible microbenchmark protocol in Appendix B). In two-stage retrieval, the dominant latency is typically incurred *after* candidate generation, when the system fetches full-precision embeddings (often from secondary storage) and computes exact similarities and optional re-ranking over the candidates (Jayaram Subramanya et al., 2019; Chen et al., 2021). By improving accuracy, QA-COS attains the same target recall with *fewer candidates*, thereby reducing downstream fetch-and-score work. Consequently, latency decreases despite the additional $\mu$s-scale per-candidate decoding cost. See Section 5.3, Appendix B, and Appendix C for native HNSW evidence, decoder-side overhead, and a sensitivity analysis of the accuracy–cost trade-off.

# 5. Empirical Evaluation in the Hard Region

**Global configuration.** We use $m$ random hyperplanes sampled i.i.d. from $\mathcal{N}(0, I_d)$ with a fixed random seed shared across methods, so that the resulting sign-only codes are directly comparable. For each database vector $A$, we store only its sign code $c_A \in \{\pm 1\}^m$ in memory, while for each query $B$ we compute query-side information (e.g., $c_B$ and $\{b_i\}$) on the fly. Throughout the main sketching experiments, the MLE is solved with damped Newton ($T$=2, $\eta$=1), setting $\varepsilon_1$=3 and $\varepsilon_2$=0.999 for numerical stability. We do not tune these constants per dataset or per baseline. All reported retrieval numbers use precomputed normalized embeddings, fixed BEIR test splits, fixed hard-query subsets selected by the baseline $\hat{\rho}_0$, and identical candidate budgets for the methods being compared. Codes will be released.

## 5.1. Angle estimation against sign-sketch baselines

**Setup.** We compare QA-COS against four sign-sketch baselines: SimHash (Charikar, 2002), SuperBit-LSH (Ji et al., 2012), Li's practical sign-full estimator $\hat{\rho}_{s,n}$ (Li, 2019), and CSSRP (Dubey et al., 2022). SimHash and QA-COS use the same random hyperplanes and the same stored database sign code $c_A$; the difference is only at decoding time. Li's $\hat{\rho}_{s,n}$ uses the same sign-full observation setting but compresses the query-side magnitudes into a reduced statistic, while QA-COS directly optimizes the full query-aware probit likelihood. SuperBit-LSH with standard orthogonal-block construction and CSSRP are included as neighboring sign-sketch baselines that change the sketch construction. We use two diagnostics: a fixed-budget variance sweep that isolates estimator stability, and a broader MAE sweep over $d \in \{256, 512, \ldots, 2048\}$ and $m \in \{256, 512, 768, 1024\}$, yielding 32 configurations after excluding near-deterministic endpoints with $|\rho| > 0.95$.

**Estimator variance over the full cosine range.** For this diagnostic, we fix $d = 256$ and $m = 128$, sweep $\rho = \cos(A, B)$, and estimate $\mathrm{Var}(\hat{\rho} - \rho)$ from $N = 2{,}000$ synthetic pairs per grid point over 10 runs (Figure 4). This targets the near-background regime where sign-only agree-

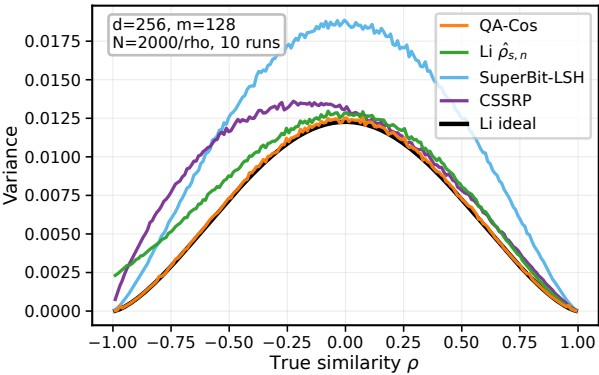

*Figure 4.* **Estimator variance over the full cosine range.** For $d = 256$ and $m = 128$ bits, we sweep the true cosine $\rho = \cos(A, B)$ and report the variance of the estimation error over 10 independent runs with $N = 2{,}000$ samples per grid point. QA-Cos uses $T = 2$ Newton steps and closely tracks Li's ideal sign-full MLE asymptotic variance, whereas SuperBit-LSH, Li's practical $\hat{\rho}_{s,n}$, and CSSRP exhibit larger variance over substantial parts of the range.

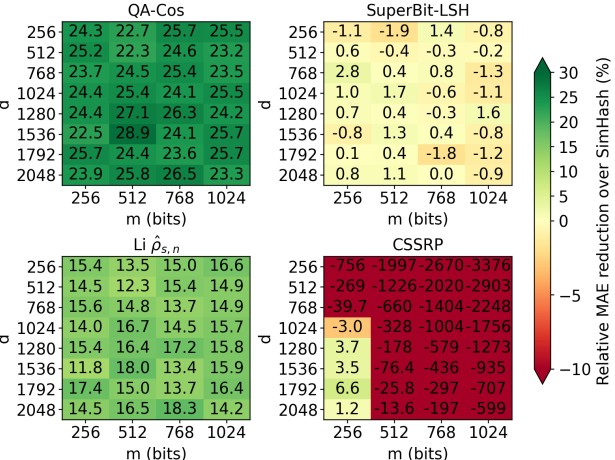

*Figure 5.* **Relative MAE reduction over SimHash across all 32 $(d, m)$ settings.** Each panel reports percentage MAE reduction relative to the standard Hamming/SimHash estimator. QA-Cos is uniformly positive across the grid, Li's $\hat{\rho}_{s,n}$ is consistently helpful but smaller, SuperBit-LSH fluctuates around zero, and CSSRP helps only in a few low-dimensional settings under this fixed-budget comparison. Values below the displayed color range are clipped in color but printed in each cell.

ment is most uncertain (Appendix G). The orange curve shows that QA-Cos nearly overlays Li's ideal sign-full MLE asymptotic variance while using the same stored database sign code as SimHash. Li's $\hat{\rho}_{s,n}$, SuperBit-LSH, and CSSRP remain above this target over substantial parts of the range, showing that the gain comes from query-aware decoding rather than a new sketch construction. The Newton solve is essentially saturated by two steps: moving from $T = 1$ to $T = 2$ reduces mean QA-Cos variance by only $0.21\%$, and additional steps change it by less than $0.0003\%$.

**Improvements across $(d, m)$ settings.** Across the 32-point $(d, m)$ sweep (Figure 5), QA-Cos reduces MAE relative to SimHash in every setting (22.3–28.9%), while Li's $\hat{\rho}_{s,n}$ gives smaller gains (11.8–18.3%). SuperBit-LSH fluctuates around zero, and CSSRP helps only in a few low-dimensional settings. Thus QA-Cos preserves the fixed sign-sketch representation while closing much of the gap to Li's ideal sign-full MLE.

### 5.2. End-to-End Retrieval and Candidate Generation

**Datasets.** Experiments are conducted on four BEIR datasets (ARGUANA, FIQA-2018, NFCORPUS, and SCIFACT) using the BEIR TEST splits with relevance judgments (QRELS) (Thakur et al., 2021). We focus on hard-query subsets, defined as the top $q \in \{0.1, 0.2, 0.3\}$ fraction of queries with the smallest $|\hat{\rho}_0|$, where $\hat{\rho}_0$ denotes the baseline sign-only estimate, corresponding to queries closest to the near-background region. This choice targets regions where the baseline estimator is least confident, rather than oracle difficulty. See Appendix H for the rationale behind dataset selection.

**Embeddings and setup.** Queries and documents are encoded using dense sentence embeddings from BEIR (Thakur et al., 2021), specifically the Sentence-Transformer model ALL-MINILM-L6-V2 (Reimers & Gurevych, 2019). All vectors are $\ell_2$-normalized and are encoded with $m=128$. To isolate decoding effects, embeddings are precomputed and cached throughout. For each query, each method sorts database items only by its decoded sketch similarity; exact dense similarities are used only for evaluation and for the downstream reranking interpretation. Thus any change in Hit@$K$ or $K$@$p$ comes from the decoder's ability to order the same binary codes, not from a different embedding model or index structure.

**Recall improvements at fixed budgets (Hit@K).** Figure 6 reports Hit@$K$ for QA-Cos and SimHash on all datasets and hard-query subsets. Hit@$K$ measures whether a qrel-relevant item survives into the top-$K$ sketch candidate list, which is the point at which a two-stage system can still recover it by exact reranking. The advantage is largest when coarse-estimation errors affect this early selection boundary: on ARGUANA HARD0.1, QA-Cos raises Hit@$K$ from 0.000 to 0.454 at $K=100$ and from 0.369 to 0.667 at $K=200$. Gains also appear on NFCORPUS (0.246 vs. 0.046 for HARD0.2 at $K=200$), SCIFACT (0.467 vs. 0.233 for HARD0.3 at $K=100$), and FIQA (0.357 vs. 0.087 for HARD0.3 at $K=200$). These improvements are strongest on near-background subsets, matching the variance analysis in Section 5.1.

**Candidate efficiency at fixed recall (K@p).** QA-Cos

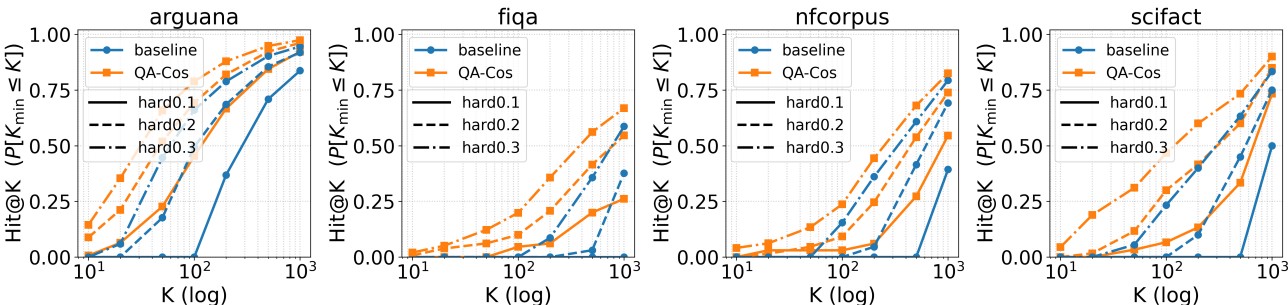

*Figure 6.* **End-to-end recall at fixed candidate budgets.** Hit@K comparing QA-COS with a sign-only baseline across four datasets. Each panel shows results for three hard-query subsets (HARD0.1, HARD0.2, HARD0.3), ordered by increasing difficulty. Across datasets, QA-COS consistently achieves higher recall than the baseline, demonstrating effective candidate selection under constrained retrieval.

also reaches fixed recall with fewer candidates (Figure 7). The $K@p$ metric reports the minimum sketch-stage candidate budget needed to reach recall level $p$, so it directly reflects how many full vectors must be fetched and rescored downstream. On ARGUANA at $p$=0.8, QA-COS reduces HARD0.1 candidates from 766 to 385 and HARD0.3 from 214 to 104; on FIQA HARD0.2, it reduces the count from 5,831 to 3,786. The effect is not simply a uniform score shift: query-aware decoding changes the local ordering near the retained frontier, where a few rank swaps determine whether a relevant item reaches the exact scoring stage.

**Implications for ANN Systems.** Together, these results show higher recall at fixed budgets and smaller candidate sets at fixed recall without modifying the database representation. This matters in two-stage and graph-based ANN systems, where approximate scores guide early selection, traversal, and pruning (Jégou et al., 2011; Johnson et al., 2017; Lin et al., 2021; Ma et al., 2023). Improving near-background estimates changes which candidates reach exact scoring, explaining the consistent end-to-end gains. The next experiments test the same claim inside a native graph implementation and then under a storage-aware wall-clock setting where reduced candidate budgets affect full-vector access time.

### 5.3. Native HNSW Scorer Replacement

**Setup.** To test whether the gain survives inside a graph ANN implementation, we patch the official hnswlib backend (Malkov & Yashunin, 2023) and replace only the search-time scorer. Insertion order, pruning, neighbor selection, level assignment, $M$, efConstruction=200, efSearch=50, and native queues are fixed. We use MP-Net embeddings on four BEIR datasets, $M \in \{16, 32\}$, 64/128-bit sketches, QA-Cos with $T = 2$, and 80 queries per dataset.

**Results.** Table 1 shows higher exact-reranked recall with fewer visited nodes. At 128 bits, Recall@10 improves from 0.770 to 0.853 and Recall@100 from 0.389 to 0.469, while

the average exact-rerank count needed for Recall@100≥0.8 drops from 479.7 to 352.2. The visited-node count also decreases (1364.5 to 1228.5 at 128 bits), indicating that better sketch scores improve both the final candidate ordering and the graph traversal path under fixed search parameters.

### 5.4. Storage-Aware Matched-Recall Latency

**Setup.** We run a focused two-stage FiQA timing experiment with the same native index, 128-bit sketches, and $M = 32$. For each scorer, we first choose the smallest efSearch from a matched-recall sweep that reaches the target exact-reranked Recall@100. The timed run then measures native HNSW search over binary sketches plus exact cosine reranking of the returned candidate IDs using normalized full-precision embeddings stored in a file-backed memmap. To make latency sensitive to candidate-set size, the full-vector store is replicated to a larger file-backed array, candidate IDs are mapped to query-dependent physical rows, and a separate file-backed buffer is touched before each timed pass to reduce repeated warm-cache hits. We use one warm-up pass and three timed passes over 80 held-out queries (240 query timings), single-threaded execution, and the same query order for both scorers. The latency-sensitive decoder is fixed in advance: QA-COS is applied only to candidates whose 128-bit agreement count lies in [74, 96], uses one Newton step ($T = 1$), and otherwise falls back to SimHash. This gate matches the failure mode studied here: it spends extra decoding only in the near-orthogonal region where SimHash is least stable and candidate selection is most margin-sensitive, rather than on high-confidence agreements near the extremes.

**Matched-recall latency.** At matched Recall@100≥0.8, Table 1 shows that the gated variant lowers efSearch from 1300 to 700, reduces the average rerank budget from 1300 to 700 full-vector fetches, and lowers mean cache-limited latency from 1.175 ms to 0.997 ms (with Recall@100 0.800 vs. 0.824). The absolute millisecond decrease is modest because FiQA is small, but the result demonstrates the

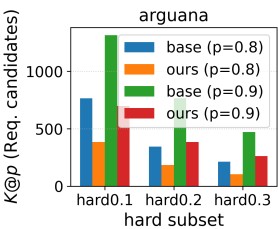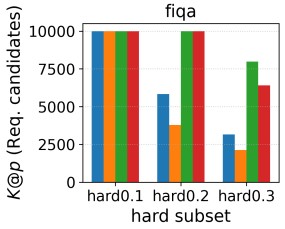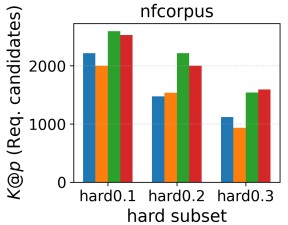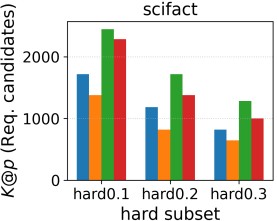

*Figure 7.* **Candidate efficiency at fixed recall** ($K@p$). Number of candidates required to achieve recall $p \in \{0.8, 0.9\}$ on hard-query subsets. Lower is better.

*Table 1.* **Native HNSW and storage-aware summaries.** Arrows show SimHash $\rightarrow$ QA-Cos. Native HNSW changes only the search-time scorer in the official `hnswlib` backend; graph construction and search parameters are fixed. Storage-aware latency uses FiQA, 128-bit sketches, $M = 32$, file-backed full-vector reranking, and the gated $T = 1$ decoder; latency is native search plus exact reranking over returned candidates, averaged over 240 query timings.

| Setting | Bits/target | Recall@10 | Recall@100 | Visited/latency | Rerank budget |
|---|---|---|---|---|---|
| Native HNSW avg. | 64 bits | .557 → .692 | .258 → .335 | 1535.5 → 1340.2 nodes | 725.4 → 566.4 for R@100≥.8 |
| Native HNSW avg. | 128 bits | .770 → .853 | .389 → .469 | 1364.5 → 1228.5 nodes | 479.7 → 352.2 for R@100≥.8 |
| FiQA, cache-limited | R@100≥.8 | – | matched | 1.175 → .997 ms | 1300 → 700 |

intended mechanism: better sketch scoring reduces both graph-search work and downstream full-vector reranking at matched recall. Decoder-side microbenchmarks and the runtime settings for the gated latency-sensitive variant are reported in Appendix B.

## 6. Related Work

Random-projection locality-sensitive hashing (LSH) is a standard route to sublinear similarity search (Indyk & Motwani, 1998; Gionis et al., 1999), with random-hyperplane hashing (SimHash) as the canonical data-independent cosine sketch (Charikar, 2002). The closest statistical precursor is Li's sign-full random projection framework (Li, 2019), which identifies a query-aware MLE but treats the full estimator mainly as a theoretical object because the estimating equation is hard to solve directly. QA-Cos makes this likelihood usable as a retrieval decoder: the $\rho = \tanh t$ reparameterization yields a smooth 1D objective, and a few damped Newton steps give a stable score while leaving database codes unchanged. We therefore compare directly with Li's practical reduced-statistic estimator $\hat{\rho}_{s,n}$, as well as other sign-random-projection improvements: additional-information estimators (Kang & Wong, 2018), related 1D MLE extensions (Kang et al., 2021), CSSRP's Count-Sketch construction (Dubey et al., 2022), and SuperBit-LSH's orthogonalized blocks (Ji et al., 2012). These comparisons separate decoder-level use of query-side information, as in Li's sign-full likelihood and QA-Cos, from sketch-construction changes such as CSSRP and SuperBit-LSH.

Broader binary-space and sketch-based methods target complementary regimes, including angular multi-index hashing and streaming multiset sketches (Qin et al., 2016; Lu et al., 2024). Data-dependent ANN systems such as PQ/OPQ, SCaNN, SOAR, and RaBitQ learn quantizers or indexing structures, often improving accuracy through training and re-indexing (Jégou et al., 2011; Ge et al., 2013; Guo et al., 2020; Sun et al., 2023; Milvus Team, 2025). Asymmetry also appears in MIPS transformations, query-adaptive hashing, and multi-probe or posterior LSH variants (Shrivastava & Li, 2014; Neyshabur & Srebro, 2015; Huang et al., 2015; Tian et al., 2022; Lv et al., 2007; Joly & Buisson, 2008; Satuluri & Parthasarathy, 2012). Unlike these representation- or index-changing approaches, QA-Cos exploits query–database asymmetry only at decoding time, improving first-stage candidate quality without retraining or index restructuring.

## 7. Conclusion

We studied cosine similarity estimation from sign-only sketches derived from random projections through the lens of query–database asymmetry, and identified the near-background region as a bottleneck of sign-only decoding. To address this, we proposed a query-aware decoder that reduces estimation error, translating into improved recall and candidate efficiency. Building on Charikar's formulation, our results show that more informative decoding is possible without changing the sketch by treating sign bits as probabilistic observations. More broadly, these findings highlight the importance of rethinking how binary sketches are decoded and demonstrate that exploiting query-time information can improve candidate generation.

## Acknowledgments

We thank the ICML 2026 reviewers for their careful reading and constructive feedback, which helped improve the clarity and empirical scope of the paper. This work was supported by the National Research Foundation (NRF), Korea, under project BK21 FOUR, and the NRF grant funded by the Korea government (MSIT) (No. 2023R1A2C2006373 and No. RS-2023-00222385).

## Impact Statement

This paper presents work whose primary goal is to advance the field of machine learning. While improvements in retrieval efficiency and accuracy may have downstream applications across a range of systems, we do not identify any immediate societal impacts that require specific discussion beyond those common to machine learning research.

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

## A. Symmetric and Asymmetric Uncertainty under Binary Sketches

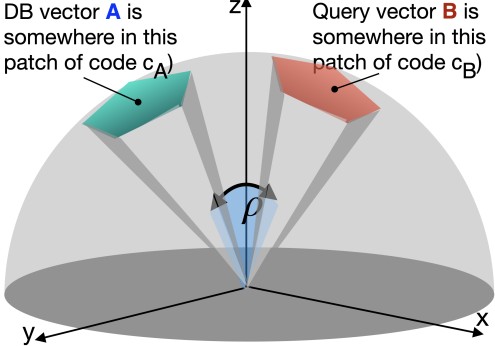

*Figure 8.* **Symmetric uncertainty under binary coding and query-side asymmetry in QA-COS.** When both the database vector $A$ and the query vector $B$ are observed only through sign-only codes, each is consistent with a feasible patch on the unit sphere (green and red regions, respectively), and the true cosine similarity is uncertain on both sides.

**Symmetric vs. asymmetric uncertainty under binary sketches.** In fully binary pipelines, both the database vector $A$ and the query vector $B$ are represented only through

*Table 2.* **Decoder-side speed/accuracy trade-off.** Runtime is mean microseconds per item on the same gated near-background candidate set used for the decoder-only microbenchmark. Error variance is reported separately as the full-$\rho$-grid average from Figure 4.

| Method | Mean runtime ($\mu s$/item) | Avg. error variance |
| --- | --- | --- |
| SimHash baseline | 0.256 | $2.894 \times 10^{-3}$ |
| Li $\hat{\rho}_{s,n}$ | 0.685 | $1.977 \times 10^{-3}$ |
| CSSRP (decode) | 0.248 | $2.197 \times 10^{-3}$ |
| QA-COS ($T = 1$) | 7.757 | $1.717 \times 10^{-3}$ |
| QA-COS ($T = 2$) | 14.971 | $1.713 \times 10^{-3}$ |
| QA-COS ($T = 6$) | 43.335 | $1.713 \times 10^{-3}$ |

sign codes $c_A$ and $c_B$. Geometrically, this implies that neither vector is uniquely specified: each is consistent with a feasible patch on the unit sphere induced by its code. As illustrated in Figure 8, the cosine similarity between $A$ and $B$ is therefore uncertain on both sides, since it depends on the unknown locations of both vectors within their respective patches.

QA-COS exploits the inherent asymmetry of two-stage retrieval pipelines: while database vectors must remain compressed for efficiency, the query vector $B$ is typically available in full precision at inference time. By retaining the real-valued query direction, QA-COS collapses the uncertainty on the query side and reduces the estimation problem to resolving the latent uncertainty induced solely by the database code $c_A$. This asymmetric view underlies the geometric interpretation in the main text and enables query-aware cosine estimation without modifying the stored codes.

## B. Decoder-Side Microbenchmarks

**What is timed.** We separate three costs that are often conflated in binary-sketch evaluation: (i) offline sketch construction, (ii) decoder-side scoring of a candidate, and (iii) downstream full-vector fetch and exact reranking. The microbenchmark in Table 2 measures only (ii), because this is the additional cost introduced by replacing Hamming decoding with a richer decoder, and pairs this cost with the corresponding estimator variance. The storage-aware experiment in Section 5.4 then reports native-search plus exact-rerank latency under matched recall. In all cases, the database sign sketches and full-precision embeddings are constructed before timing begins.

**Microbenchmark** Detailed decoder-side runtime measurements are summarized in Table 2. All experiments were conducted on a consumer-grade Apple M5 CPU (32GB RAM). We microbenchmark *decoder-side* computation only: Hamming/SimHash agreement, Li's practical sign-full estimator $\hat{\rho}_{s,n}$, CSSRP decoding, and QA-COS with Newton-based probit MLE refinement. Projection/hash

generation and all I/O are excluded. Timings condition on an identical set of *refined* candidates, defined by the near-background gate $|\widehat{p} - \frac{1}{2}| \leq 0.125$ and excluding (i) extreme-confidence cases $\min(\widehat{p}, 1 - \widehat{p}) \leq 3/m$ and (ii) tanh end-points $|\widehat{\rho}_0| \geq 0.999$. The threshold $\tau = 0.125$ corresponds to $\widehat{p} \in [0.375, 0.625]$, or approximately $|\widehat{\rho}_0| \leq 0.383$, giving a conservative band around the near-orthogonal background region where sign-only estimates are most ambiguous. For each configuration, we run multiple warm-up iterations to stabilize CPU frequency and caches, then perform 11 repeated measurements using `perf_counter_ns` and report the mean. For static decoders whose runtime does not depend on the Newton-step count, we average the means from the matched $T \in \{1, 2, 6\}$ runs. All runtime results are normalized by the number of refined items, yielding per-candidate costs in microseconds. (We fix the random seed for synthetic $(A, B)$ pairs and hyperplanes and use single-threaded BLAS to reduce variability.) The reported runtime is therefore a decoder cost for candidates that have already reached the scorer; it excludes one-time query projection, dataset loading, and downstream full-vector reranking. The variance column is not conditioned on the runtime gate: it is computed from the same estimator-variance curves used in the main text by averaging over the full $\rho$ grid.

The table makes the trade-off explicit. QA-COS is slower per refined candidate because it performs a guarded one-dimensional likelihood refinement rather than a reduced-statistic decode. However, $T = 1$ already captures almost all attainable variance reduction: moving from $T = 1$ to $T = 2$ changes the average variance only from $1.717 \times 10^{-3}$ to $1.713 \times 10^{-3}$, and $T = 6$ gives no further improvement at this precision.

**Algorithm settings used for runtime.** The default offline estimator uses the same settings as the main experiments ($T = 2, \eta = 1, \varepsilon_1 = 3, \varepsilon_2 = 0.999$). For the full native scorer-replacement study, we use $T = 2$. For the latency-sensitive storage-aware wall-clock study, we use $T = 1$ with the same-count gate [74, 96]. These choices are fixed before evaluating a dataset and are not tuned per query. The same-count gate is applied before the Newton update, and candidates outside the gate use the Hamming cosine estimate, making the runtime/accuracy trade-off deterministic for a given query code.

## C. Selective Decoding and Accuracy–Cost Sensitivity

This appendix analyzes the effect of selectively applying QA-COS decoding around the near-background region. While the main algorithm refines all non-extreme cases (i.e., $\min(\widehat{p}, 1 - \widehat{p}) > 3/m$), we additionally report a sensitivity study using a gating parameter $\tau$, which restricts decoding

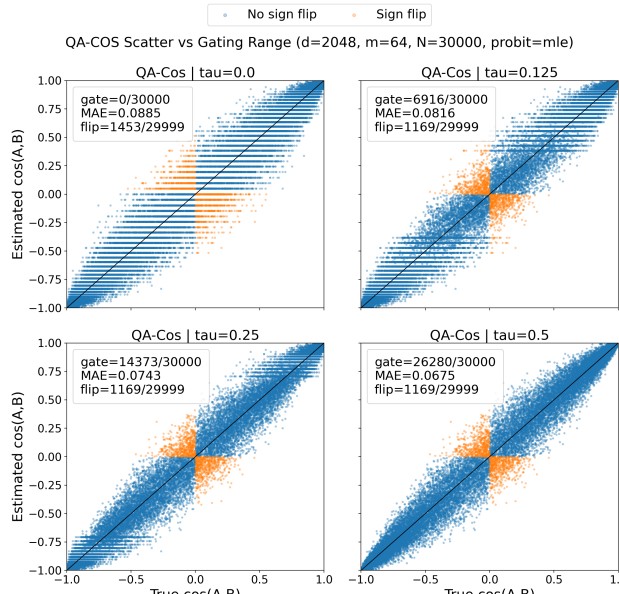

*Figure 9.* **Sensitivity analysis of selective decoding around the near-background region** (d=2,048, m=128, N=30,000, probit=MLE). The figure analyzes how selectively applying QA-Cos to samples with $|\widehat{p} - \frac{1}{2}| < \tau$ affects MAE and sign flips, while extreme-confidence cases ($\min(\widehat{p}, 1 - \widehat{p}) \leq \varepsilon_1/m$) are always excluded, and $\widehat{\rho}_0$ is clipped by $\varepsilon_2$, where $\varepsilon_1 = 3$, $\varepsilon_2 = 0.999$. Larger decoding ranges (larger $\tau$) reduce discretization artifacts in the ambiguous region and improve MAE at increased computational cost. Note that QA-COS with $\tau = 0$ reduces to the sign-only Hamming-agreement estimator. The sign-only estimate depends only on the discrete bit-agreement rate $\widehat{p} \in \{0, 1/m, \ldots, 1\}$; thus $\widehat{\rho}_0 = \cos(\pi(1 - \widehat{p}))$ takes only $m+1$ levels, yielding the horizontal banding. The configuration used in the main paper corresponds to refining all non-extreme cases.

to samples satisfying $|\widehat{p} - \frac{1}{2}| < \tau$. This analysis is included solely to illustrate the accuracy–cost trade-off and is not required for the method used in the main experiments.

**Visualization of the gating effect.** Figure 9 illustrates the effect of varying the gating range $\tau$ on QA-COS decoding (d=2,048, m=128, N=30,000, probit=MLE). As $\tau$ increases, a larger fraction of near-background samples satisfying $|\widehat{p} - \frac{1}{2}| < \tau$ is refined, while extreme-confidence cases ($\min(\widehat{p}, 1 - \widehat{p}) \leq 3/m$) remain excluded. In the figure, QA-COS with $\tau = 0$ reduces to the sign-only baseline. Increasing $\tau$ primarily affects the ambiguous similarity region around zero, reducing discretization artifacts and monotonically improving MAE (e.g., from 0.0885 at $\tau = 0$ to 0.0675 at $\tau = 0.5$), without a substantial increase in sign flips.

**Quantitative accuracy–cost trade-off.** Table 3 quantifies the relationship between the fraction of samples refined by QA-COS and the resulting MAE, based on the same near-background sensitivity analysis. Applying QA-COS to a larger subset of samples consistently improves MAE,

*Table 3.* **Accuracy–cost trade-off from selective decoder-side refinement.** Fraction of samples refined by QA-COS and the resulting MAE under different near-background ranges $\tau$ ($d$=2048, $m$=64, $N$=30,000, probit=MLE). For $\tau$=0, no refinement is applied (SimHash-equivalent); larger $\tau$ values refine a broader near-background subset. The configuration used in the main paper corresponds to refining all non-extreme cases.

| $\tau$ | Gate fraction (gated/$N$) | QA-COS MAE |
|--------|---------------------------|------------|
| 0.00   | 0.0000                    | 0.0885     |
| 0.125  | 0.2305                    | 0.0816     |
| 0.25   | 0.4791                    | 0.0743     |
| 0.50   | 0.8760                    | 0.0675     |

confirming that decoder-side refinement is most beneficial in ambiguous regions. At the same time, the gains exhibit diminishing returns: moderate refinement already captures a substantial portion of the improvement, while refining all non-extreme cases yields the lowest MAE at the highest computational cost. This analysis is included solely to illustrate the accuracy–cost trade-off; the main experiments apply QA-COS to all non-extreme cases.

**Interpretation of the sensitivity analysis.**   Taken together, Figure 9 and Table 3 illustrate how the benefits of QA-COS concentrate in the near-background region. Refining a larger subset of ambiguous cases monotonically improves accuracy, confirming that decoder-side refinement is most impactful where sign-only estimates are least reliable. At the same time, the diminishing returns observed at larger $\tau$ values indicate that a substantial portion of the gains can be achieved without refining all samples. This sensitivity analysis is included to characterize where QA-COS provides the largest benefits, rather than to introduce an additional tuning parameter in the main method.

# D. Connection to Query-Aligned Quantization in SCaNN and SOAR

This appendix clarifies the conceptual connection between the query-aware weighting principle in QA-COS and the design intuition of learned ANN systems such as SCaNN and SOAR (Guo et al., 2020; Sun et al., 2023).

**Query-aligned signal in inner-product search.**   SCaNN and SOAR are built on the observation that, for a given query, inner-product similarity is dominated by components of database vectors that are well aligned with the query, while near-orthogonal components contribute little. Accordingly, SCaNN employs anisotropic quantization strategies that prioritize preserving query-relevant projections, and SOAR further introduces redundancy to avoid missing such query-aligned signal.

**Query-aware informativeness in QA-COS.**   A closely related phenomenon arises in QA-COS. Lemma 4.1 shows that each SimHash bit is a probit observation whose informativeness about the cosine similarity $\rho$ scales with $b_i = |h_i^\top B|$. Bits with large $b_i$ contribute disproportionately to the conditional Fisher information (Theorem F.2), while bits with small $b_i$ are nearly uninformative due to near-orthogonality with the query.

**Decoder-side versus index-side realization.**   From this perspective, QA-COS and SCaNN/SOAR reflect the same underlying principle: query-aligned components dominate similarity estimation and should be emphasized. The key distinction is where this principle is realized. SCaNN and SOAR incorporate query-aligned emphasis through learned, data-dependent index structures, whereas QA-COS recovers a similar effect at decoding time from fixed binary sketches via probabilistic, query-aware weighting.

# E. Probabilistic Model for Query-Aware Decoding

We formalize the probabilistic relationship between the stored SimHash bits and the query-side projection magnitudes.

**Lemma E.1** (Bivariate normality of projections)**.** *For each random hyperplane $i$, let $a_i = h_i^\top A$ and $x_i = h_i^\top B$ denote the projections of database vector $A$ and query vector $B$, respectively, where $h_i \sim \mathcal{N}(0, I_d)$. Then $(a_i, x_i)$ is jointly Gaussian:*

$$\begin{pmatrix} a_i \\ x_i \end{pmatrix} \sim \mathcal{N}\left( \begin{pmatrix} 0 \\ 0 \end{pmatrix}, \begin{pmatrix} 1 & \rho \\ \rho & 1 \end{pmatrix} \right), \tag{8}$$

*where $\rho = \cos(A, B)$.*

*Proof.* Fix $i$ and write $h_i \sim \mathcal{N}(0, I_d)$. Define the $2 \times d$ matrix $M := \begin{pmatrix} A^\top \\ B^\top \end{pmatrix}$ so that

$$\begin{pmatrix} a_i \\ x_i \end{pmatrix} = M h_i.$$

A linear image of a multivariate normal is normal, hence $(a_i, x_i)^\top$ is Gaussian with mean $\mathbb{E}[M h_i] = M \mathbb{E}[h_i] = 0$ and covariance

$$\text{Cov}(M h_i) = M \, \text{Cov}(h_i) \, M^\top = M I_d M^\top = M M^\top.$$

Compute

$$MM^\top = \begin{pmatrix} A^\top A & A^\top B \\ B^\top A & B^\top B \end{pmatrix}$$
$$= \begin{pmatrix} \|A\|^2 & \rho \\ \rho & \|B\|^2 \end{pmatrix} = \begin{pmatrix} 1 & \rho \\ \rho & 1 \end{pmatrix},$$

using $\|A\| = \|B\| = 1$ and $\rho = A^\top B$. This proves the claim. $\qquad \square$

Conditioning on the observed query projection $x_i$ yields

$$a_i \mid x_i \sim \mathcal{N}(\rho x_i,\ 1 - \rho^2). \qquad (9)$$

Since the stored bit is $s_i = \operatorname{sign}(a_i)$, we obtain

$$\Pr[s_i = +1 \mid x_i, \rho] = \Phi\left(\frac{\rho x_i}{\sqrt{1 - \rho^2}}\right), \qquad (10)$$

where $\Phi(\cdot)$ is the standard Gaussian CDF. This probit likelihood captures how larger $b_i$ increases confidence in the observed sign, forming the basis of our query-aware maximum-likelihood estimator.

## F. Proofs for Section 4

### F.1. Geometric justification for query-aware weighting

To make the dependence on the cosine similarity $\rho$ explicit, write

$$A = \cos\theta\, B + \sin\theta\, v, \qquad v \perp B,\ \|v\|_2 = 1, \quad (11)$$

and decompose each normalized hyperplane direction $\bar{h}_i = h_i/\|h_i\|_2$ as

$$\bar{h}_i = \beta_i B + \sqrt{1 - \beta_i^2}\, u_i, \qquad u_i \perp B,\ \|u_i\|_2 = 1. \qquad (12)$$

Substituting (11)–(12) yields

$$\bar{h}_i^\top A = \beta_i \cos\theta + \sqrt{1 - \beta_i^2}\, \sin\theta\, \langle u_i, v \rangle. \qquad (13)$$

The first term is $\rho$-coupled ($\rho = \cos\theta$) and scales with $|\beta_i| = |\bar{h}_i^\top B|$, while the second term captures ambiguity from the unknown direction in $B^\perp$. Since $h_i = \|h_i\|_2 \bar{h}_i$ and $\|h_i\|_2 > 0$, this geometric alignment is consistent with the probabilistic query magnitude $b_i = |h_i^\top B|$ used by the decoder. Thus, larger query-aligned projection magnitude both amplifies the $\rho$-dependent signal and suppresses orthogonal ambiguity, increasing the Fisher information for estimating $\rho$ under the probit model.

### F.2. Proof of Lemma 4.1: Query-aware probit likelihood

*Proof.* By Lemma E.1, $(a_i, x_i)$ is jointly Gaussian with covariance $\Sigma = \begin{pmatrix} 1 & \rho \\ \rho & 1 \end{pmatrix}$. A standard conditional-Gaussian identity gives

$$a_i \mid x_i \sim \mathcal{N}\big(\mathbb{E}[a_i \mid x_i],\ \operatorname{Var}(a_i \mid x_i)\big),$$

where

$$\mathbb{E}[a_i \mid x_i] = \frac{\operatorname{Cov}(a_i, x_i)}{\operatorname{Var}(x_i)} x_i = \rho x_i,$$

$$\operatorname{Var}(a_i \mid x_i) = 1 - \frac{\operatorname{Cov}(a_i, x_i)^2}{\operatorname{Var}(x_i)} = 1 - \rho^2. \qquad (14)$$

Therefore,

$$\Pr(s_i = +1 \mid x_i, \rho) = \Pr(a_i \geq 0 \mid x_i, \rho)$$

$$= \Pr\left(\frac{a_i - \rho x_i}{\sqrt{1 - \rho^2}} \geq \frac{-\rho x_i}{\sqrt{1 - \rho^2}}\right)$$

$$= \Phi\left(\frac{\rho x_i}{\sqrt{1 - \rho^2}}\right). \qquad (15)$$

For $s_i \in \{-1, +1\}$, note that $\Pr(s_i \mid x_i, \rho) = \Pr(s_i a_i \geq 0 \mid x_i, \rho)$ and $s_i a_i \mid x_i \sim \mathcal{N}(s_i \rho x_i,\ 1 - \rho^2)$, giving

$$\Pr(s_i \mid x_i, \rho) = \Phi\left(\frac{s_i \rho x_i}{\sqrt{1 - \rho^2}}\right).$$

$\qquad \square$

### F.3. 1D MLE with stable reparameterization

**Proposition F.1** (1D log-likelihood and stable reparameterization). *Given $\{(s_i, x_i)\}_{i=1}^m$, the log-likelihood of $\rho \in (-1, 1)$ is*

$$\ell(\rho) := \sum_{i=1}^m \log \Phi\left(\frac{s_i\, \rho\, x_i}{\sqrt{1 - \rho^2}}\right). \qquad (16)$$

*With $\rho = \tanh t$ ($t \in \mathbb{R}$), this becomes*

$$\ell(t) = \sum_{i=1}^m \log \Phi\big(s_i x_i\, \sinh t\big). \qquad (17)$$

*Proof.* From Lemma 4.1, the conditional likelihood for each $i$ is

$$\Pr(s_i \mid x_i, \rho) = \Phi\left(\frac{s_i \rho x_i}{\sqrt{1 - \rho^2}}\right).$$

Assuming conditional independence across $i$ given $\rho$ (since $h_i$ are i.i.d.), the log-likelihood is the sum of log terms:

$$\ell(\rho) = \sum_{i=1}^m \log \Phi\left(\frac{s_i \rho x_i}{\sqrt{1 - \rho^2}}\right).$$

For the reparameterization $\rho = \tanh t$, we use $1 - \tanh^2 t = \operatorname{sech}^2 t$ and

$$\frac{\tanh t}{\sqrt{1 - \tanh^2 t}} = \tanh t \cdot \cosh t = \sinh t.$$

Thus $\frac{\rho}{\sqrt{1-\rho^2}} = \sinh t$ and

$$\ell(t) = \sum_{i=1}^m \log \Phi\big(s_i x_i\, \sinh t\big).$$

$\qquad \square$

Proposition F.1 reduces inference to a *one-dimensional* unconstrained optimization problem. This is a key practical advantage: the additional computation over Hamming distance is small (1D MLE), while the model can exploit the already-available query projections $\{x_i\}$.

### F.4. How does magnitude help quantitatively?

To compare estimators, we want an information-theoretic statement that makes the role of $x_i$ explicit. The Fisher information derived below shows that contributions scale with $x_i^2$.

**Theorem F.2** (Fisher information and asymptotic normality). *Define*

$$\zeta_i(\rho) := \frac{\rho x_i}{\sqrt{1-\rho^2}}, \qquad p_i(\rho) := \Phi\big(\zeta_i(\rho)\big). \qquad (18)$$

*Assume an interior true parameter $\rho \in (-1, 1)$, nonzero limiting average information, and differentiability/interchange conditions for the score. Then, under Theorem 4.3,*

$$\sqrt{m}\,(\hat{\rho} - \rho) \Rightarrow \mathcal{N}\left(0, \frac{1}{\overline{\mathcal{I}}_m(\rho)}\right), \qquad (19)$$

*where the average conditional Fisher information is*

$$\overline{\mathcal{I}}_m(\rho) = \frac{1}{m}\sum_{i=1}^{m}\mathcal{I}_i(\rho), \qquad (20)$$

$$\mathcal{I}_i(\rho) = \left(\frac{\partial \zeta_i(\rho)}{\partial \rho}\right)^2 \cdot \frac{\phi(\zeta_i(\rho))^2}{p_i(\rho)\big(1-p_i(\rho)\big)}. \qquad (21)$$

*and*

$$\frac{\partial \zeta_i(\rho)}{\partial \rho} = \frac{x_i}{(1-\rho^2)^{3/2}}. \qquad (22)$$

*Proof.* Condition on $\{x_i\}_{i=1}^{m}$. By Lemma 4.1,

$$\Pr(s_i = +1 \mid x_i, \rho) = p_i(\rho) := \Phi(\zeta_i(\rho)),$$

$$\zeta_i(\rho) := \frac{\rho x_i}{\sqrt{1-\rho^2}}.$$

Let $e_i := \mathbb{I}\{s_i = +1\} \in \{0, 1\}$. Then the $i$-th log-likelihood term is

$$\ell_i(\rho) = e_i \log p_i(\rho) + (1-e_i)\log(1-p_i(\rho)).$$

Differentiate:

$$\frac{\partial \ell_i}{\partial \rho} = \frac{e_i - p_i(\rho)}{p_i(\rho)(1-p_i(\rho))} \cdot \frac{\partial p_i(\rho)}{\partial \rho}.$$

Because $p_i(\rho) = \Phi(\zeta_i(\rho))$,

$$\frac{\partial p_i(\rho)}{\partial \rho} = \phi(\zeta_i(\rho)) \cdot \frac{\partial \zeta_i(\rho)}{\partial \rho}.$$

Compute $\partial \zeta_i(\rho)/\partial \rho$:

$$\zeta_i(\rho) = x_i \rho(1-\rho^2)^{-1/2}.$$

Let $g(\rho) = \rho(1-\rho^2)^{-1/2}$. Then

$$g'(\rho) = (1-\rho^2)^{-1/2} + \rho^2(1-\rho^2)^{-3/2}$$
$$= (1-\rho^2)^{-3/2},$$

so $\frac{\partial \zeta_i(\rho)}{\partial \rho} = x_i(1-\rho^2)^{-3/2}$.

Now compute the conditional Fisher information:

$$\mathcal{I}_i(\rho) := \mathbb{E}\left[\left(\frac{\partial \ell_i}{\partial \rho}\right)^2 \,\Big|\, \{x_i\}\right]$$

$$= \mathbb{E}\left[\frac{(e_i - p_i)^2}{p_i^2(1-p_i)^2}\left(\frac{\partial p_i}{\partial \rho}\right)^2 \,\Big|\, \{x_i\}\right]$$

$$= \frac{p_i(1-p_i)}{p_i^2(1-p_i)^2}\left(\frac{\partial p_i}{\partial \rho}\right)^2$$

$$= \frac{1}{p_i(1-p_i)}\left(\phi(\zeta_i(\rho))\frac{\partial \zeta_i(\rho)}{\partial \rho}\right)^2$$

$$= \left(\frac{\partial \zeta_i(\rho)}{\partial \rho}\right)^2 \cdot \frac{\phi(\zeta_i(\rho))^2}{p_i(1-p_i)}.$$

Independence across $i$ given $\{x_i\}$ implies additivity: $\mathcal{I}_m(\rho) = \sum_{i=1}^{m}\mathcal{I}_i(\rho)$, or equivalently $\overline{\mathcal{I}}_m(\rho) = m^{-1}\sum_{i=1}^{m}\mathcal{I}_i(\rho)$. Under the regularity assumptions stated in the theorem, classical 1D MLE asymptotic theory yields $\sqrt{m}(\hat{\rho} - \rho) \Rightarrow \mathcal{N}(0, 1/\overline{\mathcal{I}}_m(\rho))$. $\square$

Theorem F.2 makes the central effect explicit: information scales on the order of $b_i^2$. Hence, bits corresponding to large $b_i = |x_i|$ are *intrinsically more informative* about $\rho$. This provides a principled justification for query-aware weighting (implicitly done by the MLE) and suggests computational shortcuts (e.g., using only the largest-$b_i$ subset) with minimal loss.

### F.5. Proof of Theorem 4.2: information monotonicity

The following argument is a standard derivation of the Data Processing Inequality for Fisher information, included here for completeness under the regularity conditions stated in the main text. In our application, the continuous observation corresponds to the real-valued query projection $x_i = h_i^\top B$, and the coarsened observation corresponds to $y_i = g(x_i) = \text{sign}(x_i)$. Conditioning on the concurrently observed database-side sign $s_i = \text{sign}(h_i^\top A)$ yields the desired information monotonicity for $(s_i, x_i)$ versus $(s_i, y_i)$.

*Proof.* We prove the single-sample statement; the $m$-sample statement follows by additivity under i.i.d. observations.

Let $X$ have density $f_X(x; \rho)$ and $Y = g(X)$ where $g(x) = \text{sign}(x)$. Assume regularity so that the score exists and differentiation can be interchanged with integration. Define the score for $X$:

$$U(X) := \frac{\partial}{\partial \rho} \log f_X(X; \rho),$$

$$\mathbb{E}[U(X)] = 0, \quad \mathcal{I}(\rho; X) = \mathbb{E}[U(X)^2].$$

The pmf of $Y$ is $f_Y(y; \rho) = \int_{g(x)=y} f_X(x; \rho)\, dx$. Differentiate:

$$\frac{\partial}{\partial \rho} f_Y(y; \rho) = \int_{g(x)=y} \frac{\partial}{\partial \rho} f_X(x; \rho)\, \mathrm{d}x$$

$$= \int_{g(x)=y} f_X(x; \rho) U(x)\, \mathrm{d}x.$$

Therefore the score for $Y$ is

$$V(Y) := \frac{\partial}{\partial \rho} \log f_Y(Y; \rho)$$

$$= \frac{1}{f_Y(Y; \rho)} \int_{g(x)=Y} f_X(x; \rho) U(x)\, \mathrm{d}x$$

$$= \mathbb{E}[U(X) \mid Y].$$

Apply the law of total variance:

$$\text{Var}(U(X)) = \mathbb{E}[\text{Var}(U(X) \mid Y)] + \text{Var}(\mathbb{E}[U(X) \mid Y]).$$

Since $\text{Var}(U(X)) = \mathbb{E}[U(X)^2] = \mathcal{I}(\rho; X)$ and $\text{Var}(\mathbb{E}[U(X) \mid Y]) = \mathbb{E}[V(Y)^2] = \mathcal{I}(\rho; Y)$, we get

$$\mathcal{I}(\rho; X) = \mathbb{E}[\text{Var}(U(X) \mid Y)] + \mathcal{I}(\rho; Y) \ \geq \ \mathcal{I}(\rho; Y).$$

$$\mathcal{I}(\rho; s_i, X) \geq \mathcal{I}(\rho; s_i, Y)$$

$$\square$$

**Conclusion for the theorem statement.** Applying the above argument conditionally given $s_i$ (or equivalently, to the joint observation $(s_i, x_i)$ and its deterministic coarsening $(s_i, y_i)$) gives $\mathcal{I}(\rho; s_i, x_i) \geq \mathcal{I}(\rho; s_i, y_i)$; summing over $i = 1{:}m$ yields the multi-sample claim under i.i.d. observations.

### F.6. Proof of Theorem 4.3: Existence and uniqueness of the MLE

*Proof.* Let $z_i := s_i x_i$ and define $u(t) := \sinh t$, $c(t) := \cosh t$. Write $\ell(t) = \sum_{i=1}^m \log \Phi(z_i u(t))$.

**Derivative.** By the chain rule,

$$\ell'(t) = \sum_{i=1}^m \frac{\phi(z_i u(t))}{\Phi(z_i u(t))} \cdot z_i c(t) = c(t) \sum_{i=1}^m z_i \lambda(z_i u(t)), \tag{23}$$

where $\lambda(x) := \phi(x)/\Phi(x)$ is the inverse Mills ratio.

**Key property: $\lambda$ is strictly decreasing.** Differentiate $\lambda(x) = \phi(x)/\Phi(x)$ and use $\phi'(x) = -x\phi(x)$:

$$\lambda'(x) = \frac{\phi'(x)\Phi(x) - \phi(x)\Phi'(x)}{\Phi(x)^2}$$

$$= \frac{-x\phi(x)\Phi(x) - \phi(x)^2}{\Phi(x)^2}$$

$$= -\frac{\phi(x)}{\Phi(x)^2}\big(x\Phi(x) + \phi(x)\big).$$

It remains to show $x\Phi(x) + \phi(x) > 0$ for all $x$. For $x \geq 0$, $\Phi(x) \geq 1/2$ and $\phi(x) > 0$ imply positivity. For $x < 0$, integration by parts yields

$$\int_{-\infty}^x t\phi(t)\, dt = -\phi(x).$$

Since $t \leq x$ on $(-\infty, x]$,

$$x\Phi(x) = x \int_{-\infty}^x \phi(t)\, dt \ \geq \ \int_{-\infty}^x t\phi(t)\, dt = -\phi(x),$$

with strict inequality because $\phi$ has positive mass on $(-\infty, x)$ where $t < x$. Hence $x\Phi(x) + \phi(x) > 0$ and thus $\lambda'(x) < 0$.

**Existence of a maximizer.** Assume there exist $i, j$ with $z_i > 0$ and $z_j < 0$. As $t \to +\infty$, $u(t) \to +\infty$. Then for $z_j < 0$, $z_j u(t) \to -\infty$ and $\lambda(z_j u(t)) \to +\infty$, so the sum $\sum_i z_i \lambda(z_i u(t)) \to -\infty$ and $\ell'(t) \to -\infty$. Similarly, as $t \to -\infty$, $u(t) \to -\infty$ and for $z_i > 0$, $z_i u(t) \to -\infty$, so $\ell'(t) \to +\infty$. By continuity of $\ell'$, there exists $\hat{t}$ with $\ell'(\hat{t}) = 0$.

**Uniqueness.** Differentiate $\ell'(t) = c(t) \sum_i z_i \lambda(z_i u(t))$:

$$\ell''(t) = u(t) \sum_{i=1}^m z_i \lambda(z_i u(t)) + c(t)^2 \sum_{i=1}^m z_i^2 \lambda'(z_i u(t)).$$

At any stationary point $\ell'(t) = 0$, the first term vanishes (because $\sum_i z_i \lambda(z_i u(t)) = 0$ and $c(t) > 0$), hence

$$\ell''(t) = c(t)^2 \sum_{i=1}^m z_i^2 \lambda'(z_i u(t)) < 0$$

since $c(t)^2 > 0$, $z_i^2 > 0$ for all $i$ with $b_i \neq 0$, and $\lambda'(\cdot) < 0$. Therefore any stationary point is a strict local maximum. Because $\ell'(t) \to +\infty$ as $t \to -\infty$ and $\ell'(t) \to -\infty$ as $t \to +\infty$, $\ell'$ crosses zero at least once; strict negativity of $\ell''$ at any zero prevents multiple crossings. Thus the maximizer is unique. Finally, $\hat{\rho} = \tanh \hat{t}$ is unique because $\tanh$ is strictly increasing. $\square$

### F.7. Proof of Lemma 4.4: Derivatives of the query-aware probit log–likelihood

*Proof.* By the chain rule, $\frac{d}{dt} \log \Phi(z_i) = \lambda(z_i) \frac{dz_i}{dt}$ with $\frac{dz_i}{dt} = s_i x_i \cosh t$ and $\frac{d^2 z_i}{dt^2} = s_i x_i \sinh t$. Differentiating once more gives $\frac{d^2}{dt^2} \log \Phi(z_i) = \lambda'(z_i) \left(\frac{dz_i}{dt}\right)^2 + \lambda(z_i) \frac{d^2 z_i}{dt^2}$, yielding (6). $\square$

## G. Why sign-only agreement is variance-maximal near background?

**Why baseline variance matters.** The standard SimHash estimator uses only the agreement rate between sign bits of $A$ and $B$. Its behavior is governed by a Bernoulli proportion estimator, whose variance peaks at success probability $1/2$. We formalize this through the classical collision probability of random hyperplanes and a delta-method variance.

**Lemma G.1** (Random-hyperplane collision probability)**.** *Let $c_{A,i} = \text{sign}(a_i)$ and $c_{B,i} = \text{sign}(x_i)$. Then*

$$p := \Pr(c_{A,i} = c_{B,i}) = 1 - \frac{\theta}{\pi}, \qquad \theta := \arccos(\rho). \quad (24)$$

*Proof.* This is the standard random-hyperplane collision probability for angular similarity; see Charikar (2002). $\square$

Lemma G.1 connects angle estimation to estimating a Bernoulli parameter $p$ from $m$ bits. When $\theta \approx \pi/2$ (near-background), $p \approx 1/2$ and the Bernoulli variance $p(1-p)$ is maximal, which is the core reason why sign-only Hamming estimation becomes unstable in that region.

**Theorem G.2** (Delta-method variance of the Hamming-based estimator)**.** *Define $X_i = \mathbb{I}\{c_{A,i} = c_{B,i}\}$, $\hat{p} = \frac{1}{m} \sum_{i=1}^{m} X_i$, $\hat{\theta} = \pi(1 - \hat{p})$, and $\hat{\rho}_{\text{ham}} = \cos(\hat{\theta})$. For large $m$,*

$$\text{Var}(\hat{\rho}_{\text{ham}}) \approx \sin^2(\theta) \cdot \frac{\pi^2}{m} p(1-p)$$
$$= \sin^2(\theta) \cdot \frac{\pi^2}{m} \left(1 - \frac{\theta}{\pi}\right) \left(\frac{\theta}{\pi}\right), \quad (25)$$

*and at $\rho = 0$ (i.e., $\theta = \pi/2$), this variance reaches its peak:*

$$\text{Var}(\hat{\rho}_{\text{ham}}) \approx \frac{\pi^2}{4m}. \quad (26)$$

*Proof.* By Lemma G.1, $X_i$ is Bernoulli with success probability $p = 1 - \theta/\pi$, so $\text{Var}(\hat{p}) = p(1-p)/m$. Let $g(p) = \cos(\pi(1-p))$ so $\hat{\rho}_{\text{ham}} = g(\hat{p})$. Then

$$g'(p) = \frac{d}{dp} \cos(\pi(1-p)) = \pi \sin(\pi(1-p)) = \pi \sin\theta.$$

By the delta method,

$$\text{Var}(\hat{\rho}_{\text{ham}}) \approx (g'(p))^2 \, \text{Var}(\hat{p}) = \pi^2 \sin^2\theta \cdot \frac{p(1-p)}{m}.$$

Substitute $p = 1 - \theta/\pi$ to obtain the stated expression. At $\rho = 0$, $\theta = \pi/2$ gives $\text{Var}(\hat{\rho}_{\text{ham}}) \approx \pi^2/(4m)$. $\square$

Theorem G.2 formalizes the region of maximal baseline uncertainty: near $\theta = \pi/2$, the Hamming-based estimator inherits the worst-case Bernoulli variance. In a two-stage retrieval pipeline, this translates into a larger safety margin (larger candidate sets) to preserve recall. Our query-aware MLE addresses this specific weakness by leveraging the additional continuous signal $b_i$ that baseline methods discard.

## H. Dataset Selection Rationale

We evaluate QA-Cos on four BEIR datasets (ArguAna, FiQA-2018, NFCorpus, and SciFact), which together span diverse retrieval domains and operating conditions (Thakur et al., 2021). While not chosen to target a single application, these datasets collectively reflect regions where coarse retrieval quality is particularly critical.

First, the datasets differ in *semantic structure*. ArguAna and SciFact require global semantic reasoning and precise alignment, making relevance sensitive to fine-grained similarity differences. Such settings are especially vulnerable to errors in coarse similarity estimation.

Second, FiQA-2018 and NFCorpus introduce substantial *domain shift* relative to common training corpora, providing natural test cases for robustness under distributional mismatch, where data-independent sketch-based methods remain attractive (Maia et al., 2018; Boteva et al., 2016; Thakur et al., 2021).

Finally, across all four datasets, many queries operate close to the background distribution under realistic candidate budgets, either due to subtle relevance (ArguAna, SciFact) or intrinsic task difficulty and domain mismatch (FiQA-2018, NFCorpus) (Thakur et al., 2021). As a result, effective candidate generation depends critically on accurate coarse similarity estimation near the background region.

Together, these datasets form a representative testbed for studying how decoder-side improvements to binary sketch-based similarity estimation translate into end-to-end retrieval performance, independent of representation learning or re-indexing.

