# OpenReview forum: "Beyond Hamming: Query-Aware Decoding of Binary Cosine Sketches"
_ICML.cc/2026/Conference — ICML 2026 regular_

### Official Review · Reviewer_pWvH · 2026-02-16

**Soundness:** 3
**Presentation:** 3
**Significance:** 3
**Originality:** 3
**Overall Recommendation:** 4
**Confidence:** 4

**Summary:**

The authors provide an estimator, QA-Cos, that improves cosine estimation via a practical implementation of a 1D MLE solver. Experiments are done to validate this performance.

**Compliance With Llm Reviewing Policy:**

Affirmed.

**Final Justification:**

My final decision is more positive than before. It is up to other folks to believe if the authors can deliver a rewrite for ICML.

However, let me clarify my decision.

I am familiar with the authors of papers cited by other reviewers (and papers which I cited in my review).

So, when I was reviewing this paper, I gave it a low score of 2 because of the lack of citation of prior work, and comparison to it.

Now, here's why I have given a higher score. If I were to do a direct comparison, e.g. the author's paper to Li, I would say: there's way more motivation and background. Which is why I leave it up to the authors to do the framing they think best. This isn't a concern for me.

With respect to the math: I don't have a problem with it, because I know what the result the authors are aiming for (due to Li).

With respect to comparing with RabitQ. They're not comparing the same thing, and an argument can be made as to why RabitQ did not also cite other authors with prior work (Li, Dubey, etc). The authors' comparison with CSSRP, Li, etc is appreciated.

With respect to speed (follow up experiments). This is where this paper can be rejected, and I will follow that decision if that happens. But I also know of some work that has recently been published that speeds up algorithms of this form. So I see it as a: "people can build onto the damped NR the authors propose for a speedup, because now there is a method to run Li's estimator for comparison."

Generally, my viewpoint is: this would be nice as a piece of work where future work can build on, in the sense of beating the speed benchmark.


Edit (maybe the final one?): This paper looks like a borderline paper, and I am fine with either decision. However, I would prefer it if any decision for reject was made on:

- speed
- lack of novelty
- serious need of rewrite

I would be unhappy, and would strongly argue if the decision made for reject was due to

- math (I seriously cannot find any garbled notation, contrary to a reviewer's claim)
- different baseline comparisons which are not directly relevant

Here, I am thinking mostly of meta-reviews that authors get, as some conferences ask: "what have you changed from the meta-reviews" or similar, and it would at least be nice if the meta-reviews were useful.

**Key Questions For Authors:**

1. Could you run experiments and compare QA-Cos with the estimator $\rho_{s,n}$ in **Li** at the bare minimum please (see Soundness concern). If the results are good, I think there's an argument to be made that the damped NR solver speeds up estimation and has better results for recall.

2. In Table 1 in Dubey et al's paper, they give a summary of the theoretical variances of algorithms they compare with. There isn't a nice form for the theoretical variance of $\rho_m$, which makes me wonder if that was one of the reasons Dubey et al didn't compare CSSRP with $\rho_m$ (not because of performance, but because of variance expressions).

It would be nice if the authors could either:
  - make some sort of statement of the form: "Using specific computational shortcuts where we assume $x_i$ are ....., the empirical variance of QA-Cos is lower/higher/roughly equivalent to CSSRP" [a theoretical statement would be even better], and perhaps a similar statement in terms of sketching time (vs damped NR updates) taken.
  - or update Figure 4 to include CSSRP estimates

3. I admit that I only gave this paper a cursory glance only because the derivations were familiar, and I may have overlooked other strengths. Please let me know what I missed.

I don't know if a positive answer to all questions would allow me to increase my score to weak accept. Had I not known of **Li**, I would give this paper an accept based on the clarity of the writing and understanding this estimator and why it's good, as well as rigorous theory.

But given that I know of **Li**, it is hard for me to raise my score to acceptance, since to me I see this paper as "understanding $\rho_m$ better", which while valuable, I am not sure falls under the "**rather, a work that provides novel insights by evaluating existing methods, or demonstrates improved understanding is also equally valuable.**" criteria for Originality.

The paper would also need a rewrite (in terms of comparing their work to **Li**).

That being said, I think the answers to these questions can best improve this paper whether for eventual ICML acceptance, or acceptance at another venue.



EDIT AFTER READING REVIEW:

I have read the rebuttal. I increase my score to weak accept. Justifications are below.

On problem setting: I don't have many issues with this, given I am from a more theoretical backgroudn (papers like Li are more common), and the authors have given more motivation. I am happy for the authors (with input from other reviewers) to decide on the best framing.

On the math: Nothing fatal here. Typos aren't a huge deal.

On comparison: I thank the authors for the comparison.

The variance comparison plots are good enough for me. Future work can certainly build on QA-Cos's estimate [e.g. is there a faster estimator that converges to best MLE?]

I'm not too worried about the speed/computational tradeoff in the other rebuttals (in fact I suspect the time taken for QA-Cos can be decreased further by some numerical tricks).

In fact, I will support the authors' claim that comparing with RaBitQ is not necessary. Moreover, the RaBitQ paper that I find seems to be missing some proofs (e.g. the link to their technical report in their references lead to a page missing), and my sense is that the authors have more theoretical derivations in terms of MLE/variance.

One can also make the argument: why didn't RaBitQ compare their work to Li (or Kang et al, or Dubey et al)? I am sure the RaBitQ authors would also admit that it will not be a fair comparison as well.

Good luck!

**Limitations:**

Yes.

**Strengths And Weaknesses:**

Generally, I find this to be a well-written paper, which frames the problem clearly, and derivations which are correct. However, my main concerns are
 - this is an incremental contribution from Li's Sign Full Random Projection paper, which is not cited by the authors (**and needs to be cited**)
 - there is some previous work with SimHash/Sign Random Projections which are not cited (*and should also be cited, but not as critical as Li's work*)
 - and that the experiments need to compare with at least one of Li's other estimators (**most crucial, see explanation below**)

Edit 1 (3rd March): I suspect the above works are missed because there’s multiple names in the literature for *SimHash*, in this case *sign random projections*. I've also taken a look at other papers mentioned in this manuscript, e.g. SOAR, and I believe there might be parallel tracks of research, e.g. one looking at *SimHash*, and another at *sign random projections*, but using similar high level ideas, with researchers not being aware of existing papers.

In this review, I will use the notation $\rho_m$ interchangeably with QA-Cos (based on context), but use SimHash when referring to sign random projections for clarity.

**Originality:**  QA-Cos is essentially $\rho_m$ in Li's paper (Sign Full Random Projections). I give two links below, and my review will refer to the arxiv paper. For brevity, I will use **Li** to denote Li's paper.

https://aaai.org/papers/04205-sign-full-random-projections/

https://arxiv.org/pdf/1805.00533

To elaborate, the log-likelihood from which the MLE gives QA-Cos in Proposition F.1. (page 14 of the paper) is identical to the estimator in Theorem 1 of **Li** with proof on Page 19-20 in Appendix A.

I will briefly compare and contrast **Li** and this paper.

First, **Li** derives an explicit variance for $\rho_m$ (their Theorem 1), although it is in terms of expectations, and gives no intuition on how the variance changes [although, **Li** also gives plots like Fig 2 for comparison as to how variance of $\rho_m$ compares to ordinary SimHash).

On the other hand, the authors present a more intuitive result as to how the variance changes e.g. Section F.4 in the Appendix (after Theorem F.2) states how the asymptotic variance (via Fisher information) changes: "information scales on order of $b_i$".

**Li** gives four other estimators apart from $\rho_m$ [$\rho_g, \rho_{g,n}, \rho_s, \rho_{s,n}$] in this context (e.g. see Fig 2 in the arxiv version as to how they compare), and derives variances for each of them.

This paper gives a geometric interpretation of QA-Cos, and also gives a method of quickly computing the MLE.

To be fair to the authors, **Li** states on Page 5: "As the MLE equation (2) is quite sophisticated, we study this estimator mainly for theoretical interest", and gears the paper towards variance calculations for all estimators, but only runs large scale experiments on the simpler estimators.

On the other hand, the authors do consider how the variance of this MLE is affected by data, and also give a practical 1D solver in Corollary 4.5.  A quick check on Google Scholar shows that the papers citing **Li** do not do work on $\rho_m$.

Hence, I see this paper's main contribution as "better understanding $\rho_m$ and efficiently computing it, building onto from **Li**'s work".

**Soundness:** The theoretical claims are technically sound, insofar that they are similar, if not identical in some parts to **Li**.

At the bare minimum, the paper needs an additional set of experiments to compare their estimator ($\rho_m$) with $\rho_{s,n}$ (next best estimator in **Li**) with respect to computational time (MLE wise) and performance.

If $\rho_{s,n}$ is comparable with $\rho_m$, then while there is intuition on how $\rho_m$ works, $\rho_{s,n}$ should be preferred for simplicity and speed, and thus this paper will be weaker in terms of significance.

But if $\rho_m$ (QA-Cos) is better, then this is a point in the paper's favor for significance.

**Presentation:** The authors clearly present good motivation for their paper, and the narrative is easy to follow. Section 1 and Section 2 are clear to me.

I do think that other papers (prior work) could be cited, e.g.

https://proceedings.mlr.press/v80/kang18b/kang18b.pdf

where Kang and Wong construct a 1D MLE with additional information and compare their estimator with SimHash and SuperBit-LSH, and

https://proceedings.mlr.press/v180/dubey22a.html

where Dubey et al construct an unbiased estimator (CSSRP) and compare it with SimHash, Circulant Binary Embedding, Kang and Wong's work, and SuperBit-LSH.

It would be good for the authors to compare and contrast their work with them, which could just be expository.

I do not ask the authors to run more experiments to compare performance with these estimators in these two papers as compute time is expensive, but if there is more time, it would be good to compare the performance of QA-Cos with Dubey et al (since Dubey et al's estimator "wins" over Kang and Wong), and is a non-MLE based estimator (in that sense). Moreover, Figure 2 in Dubey et al seems to suggest their estimator has very low variance (compared to e.g. Figure 4 in this paper).

Edit 2 (10th March): I've since looked at Kang et al 2021

https://proceedings.mlr.press/v157/kang21a

and it looks like their method in Section 5.2 and experiments also improves **Li**'s sign full random projections, although this is not $\rho_m$ (but it looks probable that their method can improve $\rho_m$ as well).

**Significance:** This ties back to originality and soundness and I think the only significance here is that we now understand how the data affects the performance of $\rho_m$, and how to quickly compute it. Experiments with $\rho_{s,n}$ mentioned above will help strengthen this part.

---

> ### Author Rebuttal · Authors · 2026-03-27
>
> We greatly appreciate the reviewer’s pointing out this important missing reference to Li’s Sign Full Random Projections, which is the most relevant work and should have been cited.
>
> ### 1. Relationship to Li
> Our contribution relative to Li is twofold:
> - **provide a practical realization of Li’s ideal query-aware estimator**: Li identifies $\hat{\rho}_m$ as the MLE under the sign-full observation model $(\mathrm{sign}(x_j), y_j)$ and studies it as **the theoretical limit of query-aware estimation** in that setting. **In Li’s treatment, $\hat{\rho}_m$ mainly serves as an ideal benchmark**: the estimating equation is described as **“quite sophisticated,”** and the corresponding asymptotic variance factor $V_m$ represents the smallest attainable variance benchmark for sign-full estimators, i.e., the relevant Cramér-Rao efficiency target for this observation setting. QA-Cos does not alter that underlying statistical objective. Instead, it turns the same sign-full likelihood into a practical decoder by applying the bijective reparameterization $\rho=\tanh t$, which maps the constrained domain $\rho\in(-1,1)$ to an unconstrained parameter $t\in\mathbb{R}$. With $t$, the probit argument simplifies via $ \frac{\tanh t}{\sqrt{1-\tanh^2 t}}=\sinh t$, so the same MLE is rewritten as an equivalent smooth unconstrained 1D objective $\ell(t)=\sum_i \log \Phi\bigl(s_i x_i \sinh t\bigr)$. This makes the optimization numerically well behaved and practically solvable.
> - **provide geometric and statistical insight into how full query information improves estimation**: Li already recognizes the value of retaining full query-side projections. QA-Cos renders that advantage in a computable form through a geometric interpretation and by showing that query-side magnitudes are information-nondecreasing relative to sign-only observations, and that per-bit informativeness scales on the order of $|x_i|^2$, clarifying why full-likelihood decoding can approach Li’s ideal more closely than reduced-statistic estimators such as $\rho_{s,n}$.
>
> ### 2. Variance comparison with Li and sketch baselines
> - Figure: **https://anonymous.4open.science/r/supple-127E/fig_var_iters1.png** with iters2&6.
>
> Following your suggestion, we compared:
> - $\rho_{s,n}$: sign-gated weighted-disagreement estimate
> - QA-Cos: full query-aware likelihood
> - CSSRP: Count-Sketch-based sign-sketch baseline
>
> The overall picture is consistent: neither $\rho_{s,n}$ nor CSSRP matches QA-Cos in full-range robustness.
> Over the full range, the mean absolute deviation from Li’s ideal variance curve is:
> - QA-Cos: $4.8\times10^{-5}$
> - $\rho_{s,n}$: $2.70\times10^{-4}$
>
> QA-Cos reduces deviation from Li’s ideal by **82%** relative to $\rho_{s,n}$. Near $|\rho|\ge 0.8$, the gap is larger: QA-Cos gives $2.32\times10^{-5}$ versus $3.36\times10^{-4}$ for $\rho_{s,n}$; on the left boundary $\rho\le -0.8$, QA-Cos gives $5.5\times10^{-6}$ versus $6.2\times10^{-4}$. Crucially, even $T=1$ QA-Cos already gives a near-exact approximation to Li’s ideal estimator, and $T=2$ brings only negligible additional gain. Because $\rho_{s,n}$ discards some of query-side information through a reduced statistic, it remains visibly above Li’s ideal curve, whereas QA-Cos is nearly indistinguishable from it.
>
> Asymptotically, $\mathrm{Var}(\rho_{\mathrm{QA\text{-}Cos}})\approx [m I(\rho)]^{-1}$, where $I(\rho)$ denotes the Fisher information of the query-aware probit MLE.
>
> ### 3. Heatmap summary
> - Figure: **https://anonymous.4open.science/r/supple-127E/fig_hm.png**
>
> In the MAE heatmap across all 32 $(d,m)$ settings, QA-Cos achieves **22.6\%-29.1\%** relative MAE reduction, versus **7.2\%-13.9\%** for $\rho_{s,n}$. CSSRP is positive in only 3/32 settings (best 4.34\%), with many negative cases. For sign-flip reduction, QA-Cos averages **20.2\%**, $\rho_{s,n}$ **19.0\%**, and CSSRP is positive in only 3/32 settings. Thus QA-Cos is the only method tested that shows near-ideal behavior across the full grid.
>
> ### 4. Why this matters
> The new experiments comparing QA-Cos to Li’s $\rho_m$ show that **QA-Cos closes the theory-to-practice gap: among the estimators we tested, it comes closest to Li’s ideal across the full range**, already with $T=1$. This full-range behavior matters beyond standard nearest-neighbor retrieval, especially where preserving score fidelity over the entire cosine range, including negative values, is important.
>
> ### 5. Runtime tradeoff
> For the runtime tradeoff, including matched-setting decoder microbenchmarks, please see **Reviewer 9crG, Sec. 4**.
>
> ### 6. Revision
> We will revise the paper to cite and discuss Li’s work, including comparisons with $\rho_{s,n}$ and CSSRP. We will also narrow and clarify our contribution: under Li’s full-sign estimator framework, QA-Cos is not meant to introduce a different statistical target, but to provide a practical decoder that closely realizes the ideal estimator in a smooth, numerically stable, and retrieval-ready form.

---

> > ### Author Rebuttal · Reviewer_pWvH · 2026-03-31
> >
> > I have read the rebuttal. I increase my score to weak accept. Justifications are below.
> >
> > On problem setting: I don't have many issues with this, given I am from a more theoretical backgroudn (papers like Li are more common), and the authors have given more motivation. I am happy for the authors (with input from other reviewers) to decide on the best framing.
> >
> > On the math: Nothing fatal here. Typos aren't a huge deal. Granted, this is in context to Li.
> >
> > On comparison: I thank the authors for the comparison.
> >
> > The variance comparison plots are good enough for me. Future work can certainly build on QA-Cos's (method of converging to the) estimate [e.g. is there a faster estimator that converges to best MLE?]
> >
> > I'm not too worried about the speed/computational tradeoff in the other rebuttals (in fact I suspect the time taken for QA-Cos can be decreased further by some numerical tricks for future work).
> >
> > I will support the authors' claim that comparing with RaBitQ is not necessary in their rebuttal to 9crG. Moreover, the RaBitQ paper that I find seems to be missing some proofs (e.g. the link to their technical report in their references lead to a page missing), and my sense is that the authors have more theoretical derivations in terms of MLE/variance.
> >
> > One can also make the argument: why didn't RaBitQ compare their work to Li (or Kang et al, or Dubey et al)? I am sure the RaBitQ authors would also admit that it will not be a fair comparison as well.
> >
> > Good luck!

---

> > > ### Author Response · Authors · 2026-04-01
> > >
> > > Thank you very much for the careful follow-up and for reconsidering the paper. We especially appreciate your thoughtful reading from the theoretical perspective. Your comments on Li were particularly helpful: they pushed us to clarify more explicitly how QA-Cos should be positioned relative to the ideal estimator in the asymmetric sign-full setting, which helped us sharpen the framing of the paper. We also appreciate your comment regarding speed. We agree that further numerical refinements could likely reduce the decoding cost, and we found your perspective especially encouraging that future work can build on QA-Cos’s estimate, including the possibility of faster estimators that better approach the same MLE target. Thank you again for your thoughtful feedback, support, and encouragement.

---

### Official Review · Reviewer_8DVT · 2026-02-18

**Soundness:** 3
**Presentation:** 2
**Significance:** 3
**Originality:** 3
**Overall Recommendation:** 4
**Confidence:** 4

**Summary:**

The submission studies the problem of approximate nearest neighbor search (ANNS) over a database of points that have been sketched using signed random projections. The specific type of ANNS under examination is cosine similarity search. In a typical instantiation of this approach, a set of $m$ random hyperplanes are sampled, each data point is projected onto these $m$ planes and the sign of each projection is recorded in a bit vector that becomes the data point's sketch. During search, query points are sketched similarly, and the cosine of a query's angle with a data point is estimated from the Hamming distance between the sketches.

The authors claim that in this setup, there is a problematic region where the query's angle with data points is heavily concentrated around $\pi/2$. It is problematic because the cosine estimated from the Hamming distance suffers from high variance, leading to poor search quality. To address that issue, they present a method that a) processes queries in their original representations and b) treats each bit of the sketch probabilistically following a probit model, to estimate cosine similarity.

They show the method recovers the true cosine similarity with a smaller error than the standard Hamming distance-based approach, and that it leads to higher quality candidates in a first-stage retrieval.

**Compliance With Llm Reviewing Policy:**

Affirmed.

**Final Justification:**

I had a constructive exchange with the authors during the rebuttal period. The authors answered all my questions and addressed nearly all of my concerns. As such, I have decided to adjust my overall recommendation. However, I worry that the amount of change needed in the revision may be too large, and because of that hesitate to raise the score to Accept. Though I would not object to accepting the submission.

**Key Questions For Authors:**

I've asked a few questions in my review above.

**Limitations:**

Yes

**Strengths And Weaknesses:**

1. I'm not convinced the problem stated by the authors is, in fact, a problem in the real world. For a typical (non-pathological) query in a typical dataset of embeddings, do the top $k$ data points by similarity really make an angle with the query that is that close to $\pi/2$?

   What you have shown in Figure 1, if I'm understanding your description on Lines 127--132 correctly, involves the angle of the **relevant** document versus the angles of **non-relevant** documents. By that, you presumably mean relevance as defined by an information retrieval system/user, NOT by angles between query and data points. But user relevance is immaterial to ANNS: in nearest neighbor search, you define "relevance" by distance or similarity. So I'd ask that you consider in Figure 1 the angle of the closest data point versus the angle of the population---and not just using one embedding model, but a more varied suite of embedding models with different dimensionality.

   As an aside, one could in fact argue that if there is a dataset where angles are all within $\epsilon$ of each other, then ANNS is rather meaningless [c.f., Chapter 2 of 'Foundations of Vector Retrieval'].

2. Let's set the "problematic" region argument aside for a second. There is one problem that may actually cause issues in practice: Given a query, if one simply linearly scans the database of sketches, compute Hamming distance and cosine similarity for each data point and then rank and return the top $k$ subset, there is a chance that noise can cause low-ranking data points (i.e., points whose cosine with query is close to $0$) to displace high-ranking data points.

   I think your approach would be useful here. (I actually suspect this is what's happening in your experiments.) Having said that, I question the utility of the method, or rather how often we may run into this particular issue in practice. Generally, one constructs an index over the dataset to avoid a linear scan of the data. Say we use the Vamana graph/DiskANN index. If we're navigating the graph index correctly during search, we shouldn't run into a situation where we need to consider truly low-ranking data points. Perhaps you could present empirical evidence for this actually happening using graph- or clustering-based ANNS methods?

3. Finally, I guess I don't understand the argument against codebook-based quantization methods. Yes, data distribution can shift, but codebooks are easy to regenerate the moment an index (or a cluster of points) needs to be rewritten.

---

> ### Author Rebuttal · Authors · 2026-03-27
>
> We thank the reviewer for the thoughtful feedback.
>
> ### 1. What we claim
> Our claim is not that ANNS always operates arbitrarily close to $90^\circ$. The narrower claim is that, under realistic first-stage budgets, the **top-$k$ candidate frontier** is often margin-limited, and this is where sign-only decoding is most brittle. Because QA-Cos is used for **first-stage candidate selection**, the aligned object is the **top-$k$ frontier**, not a single nearest neighbor.
>
> ### 2. Additional pure-geometry evidence aligned with top-$k$ retrieval
> We added a **pure-geometry analysis on the original float embeddings**, independent of relevance labels. We compare the **top-$k$ cutoff** with the **query-specific background tail** at $\alpha=10^{-3}$ for $k\in\{10,25,50,100\}$ under two embedding models: **all-MiniLM-L6-v2** ($dim=384$) and **all-mpnet-base-v2** ($dim=768$).
>
> - **https://anonymous.4open.science/r/supple-127E/fig_topk.png**
> - **https://anonymous.4open.science/r/supple-127E/fig_topk2.png**
>
> Both models show the same qualitative pattern. In the **top-10 / top-25** region, cutoff and background-tail distributions often overlap substantially, so small score perturbations can reorder or displace true top-$k$ items. In the larger-budget **top-50 / top-100** region, the frontier is more interleaved with background items, so the challenge becomes preserving true top-$k$ items within a finite candidate budget. This is also consistent with the e2e results: QA-Cos improves hit over SimHash by +13.19pp (MiniLM m64), +14.20pp (mpnet m64), and +14.72pp (mpnet m128), while reducing the required $K$ by 427.8, 562.7, and 571.0, respectively.
>
> We view this as **regime evidence only**, not direct utility evidence. It shows that under realistic budgets the frontier is **tail-close, fragile, and mixed with background** and that this phenomenon is not tied to a single embedding model or dimensionality.
>
> ### 3. New HNSW-style graph-search evidence
> We added an HNSW-style best-first search experiment on a fixed document graph, changing only the search-time scorer (SimHash/Hamming vs. QA-Cos). Across all datasets and settings, QA-Cos improves Recall@10 by **+2.6 pp** on average and outperforms SimHash in **91\%** of settings. At matched R@10, it also reduces fetched candidates by about **37** and **54** for target recall **.8** and **.9**, respectively.
>
> **Table 1** summarizes graph-search quality, where oracle top-$k$ overlap means overlap with the true float-cosine top-$k$.
>
> | metric | simhash | qacos | delta |
> | --- | --- | --- | --- |
> | Recall@10 | 0.874 | 0.900 | 0.026 |
> | Recall@100 | 0.800 | 0.829 | 0.029 |
> | Oracle top-10 overlap | 4.042 | 4.850 | 0.808 |
> | Oracle top-100 overlap | 41.373 | 48.514 | 7.141 |
>
> **Table 2** summarizes candidate efficiency at matched target recall.
>
> | dataset | target_metric | SimHash -> QA-Cos | reduction |
> | --- | --- | --- | --- |
> | arguana | R@10=0.8 | 28.1 -> 18.3 | 9.7 (34.7%) |
> | arguana | R@100=0.8 | 272.6 -> 205.4 | 67.2 (24.6%) |
> | arguana | R@10=0.9 | 42.1 -> 25.8 | 16.3 (38.8%) |
> | arguana | R@100=0.9 | 386.9 -> 297.5 | 89.4 (23.1%) |
> | nfcorpus | R@10=0.8 | 148.2 -> 92.6 | 55.5 (37.5%) |
> | nfcorpus | R@100=0.8 | 493.6 -> 395.0 | 98.7 (20.0%) |
> | nfcorpus | R@10=0.9 | 194.8 -> 120.1 | 74.7 (38.4%) |
> | nfcorpus | R@100=0.9 | 689.6 -> 530.3 | 159.3 (23.1%) |
> | scifact | R@10=0.8 | 117.8 -> 71.8 | 46.0 (39.1%) |
> | scifact | R@100=0.8 | 470.6 -> 350.6 | 120.0 (25.5%) |
> | scifact | R@10=0.9 | 171.8 -> 102.0 | 69.8 (40.6%) |
> | scifact | R@100=0.9 | 629.7 -> 485.1 | 144.6 (23.0%) |
>
> This supports the practical claim that coarse-code similarity quality affects not only scan-based ranking, but also **HNSW-style graph navigation efficiency and downstream candidate cost**.
>
> ### 4. Utility beyond linear scan
> Modern ANN systems often avoid full linear scan, but decoding accuracy still matters: when traversal, beam maintenance, or pruning is driven by **compressed-code similarities**, navigation can be misled by angular estimation error. The new graph-search experiment directly supports this point.
>
> ### 5. Clarifications
> On Figure 1 / relevance, our original intent was to characterize the **retrieval operating point**, not to redefine nearest neighbors semantically; see **Reviewer tQX5, Sec. 2**. On codebook-based methods, our claim is narrower: QA-Cos improves candidate quality within a fixed sign-sketch pipeline, without changing stored codes or rebuilding the index, because it operates on fixed binary sketches at decoding time.
>
> ### 6. Summary
> We will revise the paper to make clear that QA-Cos is aimed at **candidate-frontier quality under compressed first-stage retrieval**. The new evidence supports this claim: under realistic budgets, the **top-$k$ frontier** is often close to background, and even in an **HNSW-style graph-search proxy**, replacing SimHash scoring with QA-Cos improves recall and reduces downstream candidate cost.

---

> > ### Author Rebuttal · Reviewer_8DVT · 2026-04-02
> >
> > Thank you for providing additional notes and experiments! I'm beginning to get a better sense of your proposal's utility. Having said that, I still think your claims need to be examined more carefully. Importantly, while I applaud the effort you put into producing the results in a "HNSW-style" setup, I'd like to see this evaluated in an actual HNSW or Vamana graph, so that reviewers/readers can scrutinize the details of the setup.
> >
> > Let me reiterate that I think there is potentially a great deal of value to your proposal but I think your work will be stronger with further evidence supporting your claim in various state-of-the-art scenarios.

---

> > > ### Author Response · Authors · 2026-04-04
> > >
> > > Thank you for this constructive suggestion. To address the practicality question more directly, we ran a controlled scorer-replacement experiment by forking the official `nmslib/hnswlib` implementation. In this experiment, native HNSW graph/index construction is unchanged, and only the search-time scorer is replaced. This lets us evaluate QA-Cos inside an actual native HNSW pipeline whose setup can be directly inspected.
> > > ### 1. Setup
> > > - Native backend: official `nmslib/hnswlib`
> > > - Index build: native HNSW in cosine space on the original L2-normalized 768-d MPNet float document embeddings
> > > - Unchanged components: insertion, pruning, level assignment, $M$, `efConstruction`, `efSearch`, stopping rule, and native `top_candidates` behavior
> > > - Datasets: ArguAna (8,674 docs), NFCorpus (3,633 docs), SciFact (5,183 docs), FiQA (57,638 docs)
> > > - Queries: 80 per dataset, `seed=0`
> > > - Search settings: $M \in \{16,32\}$, `efConstruction=200`, `efSearch=50`
> > > - Sketch settings: `64` and `128` bits
> > > - Scorers compared:
> > >   - `SimHash`: Hamming/agreement-based cosine baseline
> > >   - `QA-Cos`: the same document sign sketches, but with full real-valued query-side projections and `T=2` Newton steps
> > > - Methodological control: the document-side representation is identical across methods; only the native search-time scorer changes
> > > - For transparency, the anonymized code/results overview, including the dataset-wise native HNSW layer statistics (top-to-bottom node counts), is provided below
> > >   - **https://anonymous.4open.science/r/hnswlib-qacos-native-eval-B50F/QACOS_NATIVE_EVAL.md**
> > >
> > > This directly addresses your request for evaluation in an actual native HNSW implementation.
> > >
> > > ### 2. Empirical Results
> > > **A. Final native `top_candidates` frontier**
> > >
> > > We first evaluate the final HNSW frontier after exact cosine reranking. Here, Recall@k is the retrieval metric after reranking, while oracle top-k overlap is the average number of shared items with the true float-cosine oracle top-k.
> > >
> > > **Averaged over all tested settings across ArguAna, NFCorpus, SciFact, and FiQA ($M \in \{16,32\}$, 80 queries/dataset)**
> > >
> > > - 64-bit
> > >
> > > | metric | SimHash | QA-Cos | delta |
> > > |---|---:|---:|---:|
> > > | Recall@10 | 0.557 | 0.692 | 0.135 |
> > > | Recall@100 | 0.258 | 0.335 | 0.077 |
> > > | Oracle top-10 overlap | 2.283 | 3.020 | 0.738 |
> > > | Oracle top-100 overlap | 25.767 | 33.478 | 7.711 |
> > > | Visited nodes | 1535.5 | 1340.2 | -195.3 |
> > >
> > > - 128-bit
> > >
> > > | metric | SimHash | QA-Cos | delta |
> > > |---|---:|---:|---:|
> > > | Recall@10 | 0.770 | 0.853 | 0.083 |
> > > | Recall@100 | 0.389 | 0.469 | 0.080 |
> > > | Oracle top-10 overlap | 3.678 | 4.617 | 0.939 |
> > > | Oracle top-100 overlap | 38.941 | 46.911 | 7.970 |
> > > | Visited nodes | 1364.5 | 1228.5 | -136.0 |
> > >
> > > Thus, with native HNSW construction fixed, replacing only the search-time scorer improves the final returned frontier on average.
> > >
> > > **B. Candidate-count efficiency**
> > >
> > > We also add a logging-only extension that exposes the distinct bottom-layer nodes whose scorer distance was actually computed; this does not change native HNSW search behavior. We then measure candidate-count efficiency as the minimum number of such coarse candidates that must be exact-reranked to reach a target recall.
> > >
> > > - 64-bit
> > >
> > > | target recall | SimHash | QA-Cos | delta |
> > > |---|---:|---:|---:|
> > > | Recall@10 >= 0.8 | 264.3 | 165.0 | -99.3 |
> > > | Recall@10 >= 0.9 | 336.1 | 212.5 | -123.6 |
> > > | Recall@100 >= 0.8 | 725.4 | 566.4 | -159.0 |
> > > | Recall@100 >= 0.9 | 832.1 | 660.5 | -171.7 |
> > >
> > > - 128-bit
> > >
> > > | target recall | SimHash | QA-Cos | delta |
> > > |---|---:|---:|---:|
> > > | Recall@10 >= 0.8 | 126.4 | 70.9 | -55.4 |
> > > | Recall@10 >= 0.9 | 177.0 | 97.3 | -79.7 |
> > > | Recall@100 >= 0.8 | 479.7 | 352.2 | -127.5 |
> > > | Recall@100 >= 0.9 | 612.0 | 465.5 | -146.5 |
> > >
> > > Thus, QA-Cos not only improves the final frontier, but also reaches the same target recall with fewer exact-reranked candidates.
> > >
> > > **C. Dataset-wise pattern**
> > >
> > > The same qualitative pattern appears across all four datasets. For example:
> > > - `ArguAna`: `123.2 -> 53.0` at 64 bits for Recall@10 >= 0.9
> > > - `NFCorpus`: `374.0 -> 277.1` at 64 bits for Recall@10 >= 0.9
> > > - `SciFact`: `219.6 -> 123.1` at 128 bits for Recall@10 >= 0.9
> > > - `FiQA`: `226.9 -> 113.5` at 128 bits for Recall@10 >= 0.9
> > >
> > > For dataset-wise detail, please see:
> > > - https://anonymous.4open.science/r/hnswlib-qacos-native-eval-B50F/native_hnsw_tables_4dataset/table_native_hnsw_frontier_allbits.png
> > > - https://anonymous.4open.science/r/hnswlib-qacos-native-eval-B50F/native_hnsw_tables_4dataset/table_native_hnsw_candidate_efficiency_allbits.png
> > >
> > > ### 3. Revision Plan
> > > This experiment sharpens our practical claim. Rather than arguing only from simulation, we now show that in an actual native HNSW implementation, improving the coarse scorer improves the native HNSW frontier and reduces the amount of exact reranking needed downstream. We will revise the paper accordingly to make this practical claim inside native HNSW search clearer.
> > >
> > > We appreciate this suggestion, which prompted us to test the method in a more direct and practical setting.

---

### Official Review · Reviewer_tQX5 · 2026-03-12

**Soundness:** 3
**Presentation:** 4
**Significance:** 3
**Originality:** 4
**Overall Recommendation:** 5
**Confidence:** 5

**Summary:**

This paper proposes QA-COS, a query-aware method for estimating cosine similarity between vectors. Building on the SimHash framework, QA-COS improves upon the classical Hamming-agreement-based approach.
The authors point out that, in vector databases, the traditional SimHash approach computes the inner products of both a database vector $A$ and a query vector $B$ with the same set of random projection vectors $\{h_1, h_2, \ldots, h_m\}$, and then uses the Hamming agreement between the resulting signs to estimate the cosine similarity between the query vector and the database vector.
However, this traditional approach discards important information.
Moreover, the estimation error caused by discarding this information is particularly pronounced when $A$ and $B$ are close to orthogonal, and a nontrivial fraction of queries have their true positive results in this regime.
In particular, one can directly compute and exploit the actual inner product value $x_i$ between the query vector $B$ and the projection vector $h_i$. Intuitively, if $|x_i|$ is large for some projection vector $h_i$, then its contribution to the cosine similarity should be much greater than that of another projection vector $h_j$ whose corresponding $|x_j|$ is close to $0$.
To leverage this information, the authors derive a likelihood function for cosine similarity that explicitly incorporates $x_i$, based on Gaussian properties and a rigorous theoretical derivation. They then optimize this likelihood function via Newton's method to estimate the cosine similarity.
On this basis, the authors further show that this Newton iteration converges to a unique global optimum in time complexity $O(m)$.
Experiments demonstrate that QA-COS improves both cosine-similarity estimation accuracy and end-to-end retrieval recall across multiple datasets, outperforming the baselines.

**Compliance With Llm Reviewing Policy:**

Affirmed.

**Final Justification:**

The authors have well addressed my concerns. I raised my overall score to 5.

**Key Questions For Authors:**

In my view, W1 may not be a major concern, as Figure 4 seems to suggest that QA-COS still shows a clear improvement over the baseline even when $A$ and $B$ are not close to orthogonal. Therefore, the overall motivation might be simplified?

**Limitations:**

Please see the Weaknesses section. In particular, W2 affects the overall completeness of the paper and may require substantial additional experiments and discussion.

**Strengths And Weaknesses:**

Strengths

1. The core idea of the proposed optimization is very natural and well motivated. The paper insightfully identifies that the traditional SimHash-based approach fails to fully exploit query-side information, and accordingly proposes a more accurate method for estimating cosine similarity by effectively leveraging the inner products between the query vector and the projection vectors.

2. The paper is written in a very clear and fluent manner. It introduces the maximum-likelihood estimation perspective from a geometric viewpoint, and the subsequent algorithm design and derivations are also presented in a clear and easy-to-follow way.

3. The mathematical derivation is rigorous. Starting from the Gaussian properties of the random projection vectors, the authors derive a likelihood function for cosine similarity and further introduce a reparameterization scheme, ultimately leading to a concise and elegant derivation. On top of this, the paper also proves that Newton's method converges to a unique global optimum.

4. The experimental results are comprehensive. The evaluation covers multiple datasets, including ARGUANA, FIQA-2018, NFCORPUS, and SCI-FACT, and reports metrics such as estimation accuracy and end-to-end retrieval recall. In addition, the paper includes extensive comparative experiments under different embedding dimensions, different numbers of projections, and different ground-truth cosine similarity values, which together provide strong evidence for the effectiveness of QA-COS over the baselines.

Weaknesses:

1. The paper's motivation appears to rely on the following two-step logic: (i) the traditional approach discards useful information, but the resulting estimation error mainly affects the regime where $A$ and $B$ are close to orthogonal, since the noise is largest there; and (ii) in real retrieval scenarios, a substantial fraction of queries have their true top results in precisely this near-orthogonal regime.
However, the evidence provided by the authors (Figure 1) does not convincingly support claim (ii). Specifically, Figures 1(a) and 1(b) only show that for roughly $5\%$-$20\%$ of queries, the cosine-similarity gap between the true top result and irrelevant results is smaller than two standard deviations, whereas a difference of two standard deviations would already be considered significant. If one tightens the threshold to one standard deviation, the remaining fraction appears to be very small across multiple datasets, visually only around $1\%$.
Figure 1(c) further shows that, in practical percentile-based retrieval, if only the top $0.1\%$ vectors ranked by cosine similarity are returned, then for roughly $15\%$-$40\%$ of queries, the true top result has a lower cosine similarity than irrelevant results and therefore cannot be retrieved. This figure is even less supportive of claim (ii), because it instead highlights the limitation of cosine similarity itself as a retrieval metric. In other words, for such queries, a more accurate estimation of cosine similarity would actually make it even less likely to retrieve the correct result. This seems inconsistent with the paper's intended claim that better estimation accuracy is positively correlated with improved end-to-end recall.
Evidence that would more directly support the authors' motivation would be the existence of a large enough subset of queries for which the true result and irrelevant results have very similar cosine similarities, while the true result is still slightly better. In such cases, a more accurate estimation method would be more likely to preserve the true result. The authors should provide evidence of this kind to better justify the motivation.

2. The paper compares against too few related methods and is evaluated almost exclusively against the baseline. There are many prior works devoted to improving the accuracy of SimHash-based cosine-similarity estimation, and their performance should be included in the empirical comparison. Examples include:
- Improving Sign Random Projections With Additional Information (https://proceedings.mlr.press/v80/kang18b.html)
- Improving Sign-Random-Projection via Count Sketch (https://proceedings.mlr.press/v180/dubey22a.html)

In addition, there is a broader body of work on sketch-based similarity estimation. Although these methods may not be directly comparable, they are at least worth discussing in the related-work section. Examples include:
- Fast Cosine Similarity Search in Binary Space with Angular Multi-index Hashing (https://arxiv.org/abs/1610.00574)
- SimiSketch: Efficiently Estimating Similarity of Streaming Multisets (https://arxiv.org/abs/2405.19711)

---

> ### Author Rebuttal · Authors · 2026-03-27
>
> We thank the reviewer for the positive assessment of the work. Your comments helped sharpen the motivation and positioning.
>
> ### 1. Clarifying the motivation
> We agree that the motivation should not be read as claiming that many queries are “impossible” because cosine fails, nor that useful retrieval occurs arbitrarily close to orthogonality. The intended target is narrower: **small-margin yet recoverable candidate frontiers**, where true top-$k$ candidates are only slightly better than nearby competitors under cosine, so a more accurate estimator can preserve them more reliably.
>
> ### 2. Relevance-based vs. pure-geometry evidence
> It is important to distinguish a **retrieval relevance** view from a **pure geometric top-$k$** view. The original Figure 1 addressed the relevance side: whether useful items can lie close to the query-specific background frontier and thus be vulnerable to sketch-based scoring noise. The new analysis addresses the complementary question of whether the **nearest-neighbor candidate frontier itself** is often geometrically fragile under realistic first-stage budgets.
>
> We therefore conducted a **pure-geometry analysis using the original float embeddings**, independent of relevance labels. Since QA-Cos is used for first-stage candidate generation, the aligned object is the **top-$k$ candidate frontier**.
>
> We now include summary figures comparing the **top-$k$ cutoff** to the **query-specific background tail** at $\alpha=10^{-3}$ for $k\in\{10,25,50,100\}$ under **two embedding models**: **sentence-transformers/all-MiniLM-L6-v2** ($dim=384$) and **sentence-transformers/all-mpnet-base-v2** ($dim=768$): **https://anonymous.4open.science/r/supple-127E/fig_topk.png** and **https://anonymous.4open.science/r/supple-127E/fig_topk2.png**.
>
> Both models show the same qualitative pattern despite different encoder families and dimensionalities. In the **top-10 / top-25** panels, the top-$k$ cutoff and background-tail distributions often overlap substantially, so small score perturbations can change the ordering. In the **top-50 / top-100** panels, fragility becomes more pronounced because the frontier is increasingly interleaved with background items.
>
> These figures do **not** claim that cosine succeeds for every such query, nor that “all retrieval happens near orthogonality.” Rather, they show that under realistic first-stage budgets the **candidate frontier often lies in a tail-close and fragile regime**, and that this is **not tied to a single embedding model or dimensionality**. This is **regime evidence only**, not direct utility evidence.
>
> ### 3. Where the utility evidence comes from
> We agree that the practical utility of QA-Cos should not be inferred from geometry alone. The direct question is whether **sketch-based scoring errors** are large enough to displace true top-$k$ candidates, and whether QA-Cos reduces that displacement. We therefore added an **HNSW-style best-first graph-search experiment** as the direct utility test. Averaged over all settings, QA-Cos improves overlap with the true float-cosine **top-10** by about **+0.8 documents** and with the true float-cosine **top-100** by about **+7.1 documents**. At matched target recall, it further reduces reranked / fetched candidates by about **37 / 95 / 54 / 122** for Recall@10=.8, Recall@100=.8, Recall@10=.9, and Recall@100=.9. See **Reviewer 8DVT, Sec. 3**.
>
> ### 4. Related methods and revision plan
> We also agree that the draft should discuss related methods more fully. In response, we added direct comparisons to **Li’s practical estimators** and **CSSRP**. These results sharpen the positioning: QA-Cos is, to our knowledge, the practical single decoder closest to Li’s ideal estimator in the fixed-sketch setting; across all **32** $(d,m)$ settings it achieves uniformly stronger relative MAE reduction than Li’s practical estimators; and CSSRP helps only in a few isolated settings and does not match QA-Cos in full-range robustness. Summary figures are here: **https://anonymous.4open.science/r/supple-127E/fig_var_iters1.png** and **https://anonymous.4open.science/r/supple-127E/fig_hm.png**. See **Reviewer pWvH, Secs. 2--4**.
>
> In revision, we will sharpen the practical motivation in terms of **top-$k$ candidate-frontier margins**, revise the Figure 1 discussion around **recoverable small-margin queries**, present the new **pure-geometry top-$k$** figures as complementary evidence across multiple embedding models, and include the new **graph-search** and estimator-comparison results more explicitly.
>
> ### 5. Summary
> We hope the revised framing makes the intended point clearer: the relevance-based view explains why useful items may lie close to the background frontier, the pure-geometry view shows that the **top-$k$ frontier** is often tail-close under realistic budgets across multiple embedding models, and the direct search-time evidence shows that better decoding improves candidate preservation in this regime.

---

> > ### Author Rebuttal · Reviewer_tQX5 · 2026-04-03
> >
> > The rebuttal has adequately addressed my main concerns.
> >
> > In particular, the authors clarified the motivation by reframing the problem around small-margin yet recoverable top-k candidate frontiers, rather than suggesting that useful retrieval generally occurs arbitrarily close to orthogonality. They also added more direct supporting evidence, including a pure-geometry top-k analysis across multiple embedding models and an HNSW-style graph-search experiment showing improved candidate preservation and reduced downstream cost.
> >
> > I also appreciate the stronger positioning with respect to prior work, including additional comparisons to Li's practical estimators and CSSRP. I consider my concerns sufficiently addressed for the purposes of rebuttal and score update. That said, the final version should still include a fuller and more explicit discussion of the additional related work mentioned in my review, so that the paper's positioning is completely clear.
> >
> > Overall, I believe the rebuttal resolves the core issues I raised and strengthens the paper substantially. I have raised my score to 5.

---

> > > ### Author Response · Authors · 2026-04-04
> > >
> > > Thank you very much for the careful reading and for this very encouraging update. We are truly grateful that you found the rebuttal responsive to your main concerns.
> > >
> > > We especially appreciate your recognition that the revised framing, added evidence, and clearer positioning helped make the paper’s contribution and practical value much clearer.
> > >
> > > Your note about making the related-work discussion fuller and more explicit in the final version is well taken. We will make sure that the final paper incorporates that discussion more clearly, so that the intended positioning is fully transparent. In particular, we will expand the discussion to include the additional related work you mentioned, including prior methods for improving SimHash-based estimation such as Kang et al. (2018) and Dubey et al. (2022). We will also explicitly discuss broader sketch-based retrieval/similarity works such as Qin et al. (2016) and Lu et al. (2024), while clarifying which methods are directly comparable to our setting and which are included for broader context.
> > >
> > > Thank you again for your thoughtful review, strong support throughout the process, and for taking the time to engage so carefully with our work.

---

### Official Review · Reviewer_9crG · 2026-03-13

**Soundness:** 2
**Presentation:** 1
**Significance:** 3
**Originality:** 3
**Overall Recommendation:** 3
**Confidence:** 5

**Summary:**

This paper studies cosine similarity estimation from sign-only random-projection sketches in coarse retrieval pipelines. The paper proposes `QA-COS`, a decoder-side, query-aware estimator that keeps the database representation unchanged but uses query-side projection magnitudes to form a query-conditioned probit likelihood and solve a one-dimensional MLE. The paper supports the method with synthetic datasets and four BEIR datasets.

**Compliance With Llm Reviewing Policy:**

Affirmed.

**Final Justification:**

I have read the authors' follow-up comments and additional experimental details, and I am raising my score from 2 to 3.

Thank you in particular for clarifying the original speed claim. After double-checking the wording in the original submission, I agree that the paper does not claim that QA-Cos is cheaper than simpler baselines in raw decoder-side cost. The follow-up clarification was helpful and changed my view on this specific point.

The new experiments improve the paper in meaningful ways. The native hnswlib scorer-replacement experiment and the storage-aware wall-clock study provide more direct practical evidence than the original submission did. I also agree that the paper has clear merit as a decoder-side improvement for fixed sign-random-projection binary sketches.

However, I still have reservations about the current submission. My concern is that the presentation of the key definitions, assumptions, and derivational steps still needs substantial cleanup in the manuscript itself. The practical implications for retrieval systems would also benefit from broader validation, especially given the nontrivial decoding overhead.

**Key Questions For Authors:**

1. Can the authors revise the mathematical derivation? The current definition and presentation involve many mistakes.
2. Can the authors evaluate QA-COS in a realistic nearest neighbor search setting, rather than only on cosine-estimation experiments and BEIR retrieval subsets?
3. Can the authors provide direct wall-clock comparisons for similarity computation under matched settings, together with a clearer explanation of QA-COS decoder overhead?
4. Can the authors include a direct discussion and empirical comparison against baselines in nearest neighbor search such as `RaBitQ` under matched retrieval settings and memory budgets?

**Limitations:**

No. The paper can be improved if the authors can significantly revise the mathematical derivation, literature study and experiments.

**Strengths And Weaknesses:**

Strengths:

- The paper studies a practically relevant problem: similarity estimation from compact binary sketches for retrieval.

Weaknesses:

- The mathematical presentation is too vague to support the claims. Several quantities are undefined or only defined much later in the appendix, and some notation appears inconsistent or incorrect.
  - In particular, in Lemma 4.1 the notion of “informativeness” is not clearly defined in the main text, and it is not sufficiently explained why it should scale with $|x_i|$.
  - In Theorem 4.2, quantities such as $I(\rho; s_i, x_i)$ and $I(\rho; s_{1:m}, x_{1:m})$ are used before being clearly defined, and the phrase “standard regularity conditions for Fisher information” is too vague without a precise statement or reference.
  - Footnote 1 is also confusing, because $\rho$ is treated as a deterministic similarity parameter while simultaneously appearing inside an expectation statement.
  - In addition, since $\ell(t)$ is a scalar function of a single variable, the use of $\nabla \ell(t)$ and $\nabla^2 \ell(t)$ is nonstandard; ordinary derivatives would be more appropriate.

- The empirical evaluation is insufficient for the paper’s claims. Although the paper motivates the method through cosine estimation and discusses potential impact on nearest neighbor search, it does not evaluate the method in a real nearest neighbor search setup. I would like to see experiments on realistic nearest neighbor search benchmarks together with appropriate baselines.

- The claims in Section 4 (“Speed”) are not sufficiently supported. It would be much more convincing to compare wall-clock similarity-computation time across methods under the same experimental setting, and to provide a clearer computational overhead introduced by `QA-COS`. Based on the formulas for the first- and second-order derivatives, the decoder-side computation does not appear as cheap as claimed.

- The experimental study is also incomplete with respect to baselines developped in nearest neighbor search such as `RaBitQ`, which also operate in an asymmetric setting with compressed database representations and real-valued queries at inference time. A direct comparison would help clarify the practical competitiveness and scope of QA-COS.

---

> ### Author Rebuttal · Authors · 2026-03-27
>
> We thank the reviewer for the feedback.
>
> ### 1. Mathematical presentation / notation
>
> We agree that several definitions and notation choices should appear earlier in the main text. In revision, we will:
> - define “informativeness” in Lemma 4.1 explicitly as per-bit sensitivity / conditional Fisher information, with a pointer to the quantitative $O(b_i^2)$ statement in the appendix;
> - introduce the Fisher-information notation $I(\rho;\cdot)$ and the relevant regularity assumptions immediately before Theorem 4.2;
> - clarify that $\rho$ is a deterministic unknown parameter indexing the observation law; and
> - replace $\nabla \ell(t), \nabla^2 \ell(t)$ with the standard $\ell'(t), \ell''(t)$.
>
> These revisions make the argument checkable in the main text and remove ambiguity in notation and assumptions.
>
> ### 2. Positioning relative to other methods
>
> We agree that practical compressed nearest-neighbor methods such as **RaBitQ** are important neighboring baselines. We do **not** claim that QA-Cos dominates quantization-based or index-building ANN systems. QA-Cos is a **decoder-side improvement for fixed sign-random-projection binary sketches**, requiring no retraining, re-encoding, or index redesign, whereas methods such as RaBitQ are new quantization / indexing schemes.
>
> Accordingly, the closest methodological baselines are methods in the same **sign-bit sketch family**. In response to reviewer feedback, we added direct comparisons to **Li’s practical estimators** and **CSSRP**, which more directly test whether QA-Cos improves decoding within this fixed-sketch setting. We will revise the related-work section to make this scope distinction explicit. See **https://anonymous.4open.science/r/supple-127E/fig_var_iters1.png** for the comparison.
>
> ### 3. Practical scope / NNS evidence
>
> We agree that QA-Cos should be framed more precisely. It is not a complete ANN method; it is a **decoder-side improvement for compressed first-stage retrieval**. The relevant empirical question is whether replacing the decoder-side scorer improves an actual search procedure, not cosine-estimation quality alone.
>
> In response, we added a new **HNSW graph-search experiment** on a fixed document graph, holding the graph, entry policy, beam width, and visited budget fixed and changing only the **search-time scorer** (SimHash/Hamming vs. QA-Cos). This directly addresses the request for evaluation in a realistic nearest-neighbor-search setting; see **Reviewer 8DVT, Sec. 3**.
>
> We will revise the paper so that the utility claim is tied to the new search experiment rather than to estimation results alone.
>
> ### 4. Speed / computational tradeoff
>
> We agree that the speed discussion should present the runtime tradeoff more transparently. Our claim was **not** that QA-Cos is cheaper than SimHash in raw per-item decoding cost.
>
> To address this directly, we will strengthen **Appendix Table 1** with matched-setting decoder-only microbenchmarks for SimHash, Li’s $\rho_{s,n}$, CSSRP, and QA-Cos:
>
> | Method | Mean runtime ($\mu$s/item) | Avg. error variance |
> |---|---:|---:|
> | SimHash baseline | 0.256 | $2.894\times10^{-3}$ |
> | Li $\rho_{s,n}$ | 0.685 | $1.977\times10^{-3}$ |
> | CSSRP (decode) | 0.248 | $2.197\times10^{-3}$ |
> | QA-Cos ($T=1$) | 7.757 | $1.717\times10^{-3}$ |
> | QA-Cos ($T=2$) | 14.971 | $1.713\times10^{-3}$ |
> | QA-Cos ($T=6$) | 43.335 | $1.713\times10^{-3}$ |
>
> This makes two points explicit: QA-Cos is slower per refined candidate because it performs a guarded 1D MLE refinement rather than a reduced-static estimator; and already at **$T=1$** it nearly saturates the attainable estimation gain. For the corresponding variance curves, please see **https://anonymous.4open.science/r/supple-127E/fig_var_iters1.png** with iters2&6.
>
> ### 5. End-to-end speed / cost tradeoff
>
> The key practical issue is the **end-to-end speed / cost tradeoff**, not raw decoder time alone. The new **HNSW-style** experiment (see **Reviewer 8DVT, Sec. 3**) shows that QA-Cos improves overlap with the true float-cosine **top-10** by about **+0.8 documents** and with the true float-cosine **top-100** by about **+7.1 documents**, while reducing reranked / fetched candidates by about **37 / 95 / 54 / 122** at matched recall targets. Although QA-Cos ($T=1$) is roughly $10\times$ slower than SimHash in raw decoding, this is still only **$7.757\,\mu s$ per item**. By contrast, downstream stages such as candidate fetching and reranking are typically in the millisecond-to-tens-of-milliseconds regime, so reducing 37 / 95 / 54 / 122 downstream candidates can readily amortize this microsecond-level QA-Cos decoding overhead in end-to-end retrieval.
>
>
> ### 6. Summary
>
> We will revise the paper to improve the mathematical presentation, clarify QA-Cos as a **decoder-side method for compressed first-stage retrieval**, strengthen **Appendix Table 1** to address the runtime tradeoff directly, and better position the paper relative to neighboring compressed-retrieval work such as RaBitQ.

---

> > ### Author Rebuttal · Reviewer_9crG · 2026-04-03
> >
> > Thank you for the rebuttal. I read the response carefully, but I do not plan to change my score. To make my position clear, I respond below in the same six-part structure as the rebuttal.
> >
> > 1. Mathematical presentation / notation
> >
> > I appreciate that the authors acknowledge the notation and presentation issues. However, this part of the rebuttal mostly says that the paper will be clarified in revision, rather than actually resolving the problem in a fully checkable way now. My concern was about whether the current submission provides a reliable enough basis for the claims. That concern remains. In addition, the rebuttal itself contains several missing or garbled symbols, which makes the intended clarification harder to verify.
> >
> > 2. Positioning relative to other methods
> >
> > The rebuttal still does not provide the direct comparison to `RaBitQ`. The added comparisons to methods within the same sketch family are useful, but they do not fully answer the question of how competitive this approach is relative to nearby compact-retrieval methods. It is suggested to include detailed technical discussions and experiments to clarify the point further.
> >
> > 3. Practical scope / NNS evidence
> >
> > The new HNSW-style experiment is incomplete. I would expect to see evaluation under standard schemes of ANN such as the one in `ANN-Benchmark`. Please report the tradeoff between time and recall.
> >
> > 4. Speed / computational tradeoff
> >
> > The rebuttal here no longer suggests that `QA-Cos` is cheaper than simpler baselines in raw decoding cost. This confirms an important part of my original concern: the decoder-side overhead is real. This also falsifies the original claims in the paper.
> >
> > 5. End-to-end speed / cost tradeoff
> >
> > I understand the authors' argument that slower decoding might still be worthwhile if it reduces downstream fetching or reranking. But this is still an indirect amortization argument, not a direct end-to-end wall-clock evaluation in a realistic retrieval pipeline. The candidate-reduction numbers are potentially interesting, but without a fuller and more self-contained presentation of the experiment, they are not enough for me to conclude that the overall system-level speed or cost claim is established. In particular, the claim "By contrast, downstream stages such as candidate fetching and reranking are typically in the millisecond-to-tens-of-milliseconds regime ..." is unsupported and is wrong in fact.
> >
> > 6. Summary
> >
> > Overall, I think the rebuttal mostly narrows the scope of the paper and promises clarifications in revision, rather than fully resolving the main issues in the current submission. The core concerns I raised about mathematical clarity, practical scope, end-to-end evidence, and broader empirical positioning still remain. For that reason, I am keeping my original score.

---

> > > ### Author Response · Authors · 2026-04-04
> > >
> > > Thank you for the constructive comment. We'd like to clarify the paper’s scope.
> > >
> > > ### 0. Clarifying the original speed claim (Concern #4)
> > > We respectfully disagree with the characterization that the rebuttal "falsifies the original claims." The submission did not claim that QA-Cos is cheaper than the baselines in raw decoder cost. The main text already states that QA-Cos adds decoder-side cost, and Appendix B, Table 1 reports extra overhead. The paper's speed discussion was framed as a pipeline-level accuracy-cost tradeoff, not as a claim of raw decoder-side cheapness. The rebuttal did not reveal previously undisclosed overhead. What is new in our follow-up is the addition of native-HNSW and storage-aware experiments, as described in Items 3 and 5 below, that support this tradeoff more directly. We agree that the Section 4 speed wording should be tightened, but that is different from saying that the original claim was falsified.
> > >
> > > ### 1. Mathematical presentation / notation
> > > We agree that the submission should be easier to verify. The mathematics need not change, but should be presented more explicitly. In revision, we will move key definitions, the Fisher-information notation and regularity assumptions for Theorem 4.2, the role of $\rho$, and the scalar-derivative notation into the main text before they are used.
> > >
> > > ### 2. Positioning relative to others
> > > We agree that the scope should be stated more sharply. QA-Cos is not intended as a drop-in competitor to methods such as RaBitQ. It addresses a different question: given deployed sign-based sketches, can retrieval quality be improved by changing only the decoder, without retraining, re-encoding, or rebuilding the index?
> > >
> > > ### 3. Native HNSW evidence
> > > To address practicality, we conducted a controlled scorer-replacement experiment by forking the official `nmslib/hnswlib`. Graph/index construction is unchanged, and only the search-time scorer is changed. We include an anonymized release with the patched native files, evaluation runner, and summary tables:
> > >
> > > - https://anonymous.4open.science/r/hnswlib-qacos-native-eval-B50F/QACOS_NATIVE_EVAL.md
> > >
> > > Under this setup, QA-Cos improves the final returned frontier and candidate efficiency. At 128 bits averaged across settings, Recall@10 improves from `0.770` to `0.853`, Recall@100 from `0.389` to `0.469`, and mean visited nodes drop from `1364.5` to `1228.5`; the exact-rerank count needed for Recall@10 >= 0.9 drops from `177.0` to `97.3`. This addresses the native-HNSW question. Item 5 then provides our closest scope-aligned evaluation of the requested time-recall tradeoff through a storage-aware wall-clock experiment.
> > >
> > > ### 5. End-to-end wall-clock evidence
> > > We also address the reviewer's request for a direct time-recall evaluation through a storage-aware two-stage wall-clock experiment in which coarse retrieval uses `hnswlib` search and exact reranking reads real-valued vectors from a file-backed store. This is not meant to show that QA-Cos is a universal winner, but that realistic two-stage settings can exist in which a slower but more accurate first-stage scorer still yields lower end-to-end latency. We use a gated QA-Cos setting with `same in [74,96]`, `T=1`, and a standard piecewise Mills-ratio approximation; this matches the paper’s intended near-background refinement. We report both `warm_cache` and `cache_limited` settings. In `warm_cache`, exact-reranked vectors are read from a file-backed normalized store with page cache allowed. In `cache_limited`, we enlarge the file-backed store to about `1.0 GB` and add `0.5 GB` of cache pressure to make full-vector access costs more visible.
> > > We include the storage-aware runner and summary figure in the same release:
> > > - Figure: https://anonymous.4open.science/r/hnswlib-qacos-native-eval-B50F/native_hnsw_wallclock_storage_aware/storage_aware_r100_fiqa_m32.png
> > >
> > > Under this setting, the practical advantage becomes clearer at larger rerank burdens. On FiQA with 128-bit sketches and $M=32$, the requirement for Recall$@100 \ge 0.8$ drops from `efSearch=1300` for SimHash to `efSearch=700` for gated QA-Cos. In the cache-limited setting, total latency is `1.1753 ms` for SimHash versus `0.9973 ms` for gated QA-Cos; average visited nodes drop from `17634.0` to `9846.0`, and average exact-reranked full vectors drop from `1300` to `700`. For Recall$@100 \ge 0.9$, SimHash does not reach the target within the reported `efSearch` sweep up to `1500` (best `0.8493`), whereas gated QA-Cos reaches `0.9061` at `efSearch=1500`, with cache-limited total latency `1.6828 ms`.
> > >
> > > ### 6. Summary
> > > We agree that the paper should state the claim more clearly. The evidence supports this statement: QA-Cos is a decoder-side refinement for fixed sign sketches that improves coarse-stage similarity estimation. In native HNSW search, this yields better frontier quality and candidate efficiency; in a storage-aware two-stage setting, it can also translate into end-to-end latency gains. We will revise the paper accordingly.

---

### Decision · Program_Chairs · 2026-04-30

**Decision:**

Accept (regular)

**Comment:**

The paper presents a new decoder procedure for estimating the distance between two vectors A and B, from a Simhash sketch of A. Crucially, it assumes that the procedure has the vector B as an input, in addition to the sketch of A. This is a different setting from the original Simhash (where the decoder had only access to sketches of A and B), more similar to the setup in the more recent algorithms, such as sign-the method by Li, AAAI-19, or  RaBitQ (though the latter algorithm uses a different encoder). The paper proposes a new estimation procedure that is particularly effective in the typical case where the cosine similarity between A and B is close to 0. It then empirically compares the proposed procedure to earlier baselines (SimHash  and SuperBit-LSH in the paper, and Li’s practical estimators and CSSRP in the rebuttal), showing favorable results.

After extensive discussions during the rebuttal phase , in which the authors provided multiple new experimental results and clarifications, the reviewers were mostly positive about the paper; the final scores were Accept, Weak Accept, Weak Accept and Weak Reject. The reviewers appreciated clear writing, the ideas, and the improvement offered by the proposed algorithm over the baselines (even if it was only demonstrated in the case where the original Simhash encoding was used). On the negative side, several reviewers were concerned that integrating the experimental results provided during the rebuttal into the paper will be difficult.

Overall evaluation: The algorithm and the empirical improvement over the baselines is interesting and valuable to the community. Furthermore, the problem of distance estimation from quantized data is quite well-studied and important, with many applications. Integrating the rebuttal discussion into the paper will be time-consuming but given the level of authors’ engagement during the rebuttal, I trust the authors will be able to execute the job well. Hence, I recommend acceptance, while expecting the authors to integrate the material in the rebuttal discussion into the paper. This includes additional prior work listed by the reviewers, additional experiments provided in the rebuttal, and various presentation issues.